# Transport mechanism of DgoT, a bacterial homolog of SLC17 organic anion transporters

Natalia Dmitrieva [ID] [1], Samira Gholami [ID] [1,2], Claudia Alleva [ID] [1,7], Paolo Carloni [ID] [2,3,4,5], Mercedes Alfonso-Prieto [ID] [2,6,8] & Christoph Fahlke [ID] [1,8 ✉]

## Abstract

The solute carrier 17 (SLC17) family contains anion transporters that accumulate neurotransmitters in secretory vesicles, remove carboxylated monosaccharides from lysosomes, or extrude organic anions from the kidneys and liver. We combined classical molecular dynamics simulations, Markov state modeling and hybrid first principles quantum mechanical/classical mechanical (QM/MM) simulations with experimental approaches to describe the transport mechanisms of a model bacterial protein, the D-galactonate transporter DgoT, at atomic resolution. We found that protonation of D46 and E133 precedes galactonate binding and that substrate binding induces closure of the extracellular gate, with the conserved R47 coupling substrate binding to transmembrane helix movement. After isomerization to an inward-facing conformation, deprotonation of E133 and subsequent proton transfer from D46 to E133 opens the intracellular gate and permits galactonate dissociation either in its unprotonated form or after proton transfer from E133. After release of the second proton, *apo* DgoT returns to the outward-facing conformation. Our results provide a framework to understand how various SLC17 transport functions with distinct transport stoichiometries can be attained through subtle variations in proton and substrate binding/unbinding.

**Keywords** Major Facilitator Superfamily; Molecular Dynamics Simulations; Organic Anion Transporter; Proton-coupled Secondary Transport; Solid-supported Membrane Electrophysiology
**Subject Category** Structural Biology

## Introduction

Solute carrier 17 (SLC17) transporters fulfill a variety of cellular functions (Reimer, 2013; Omote et al, 2016; Li et al, 2022). They transport diverse anionic substrates, utilizing mostly electrochemical proton (H+) gradients as the driving force, with a large variability in transport stoichiometry ranging from electrogenic H+-glutamate exchange by VGLUT (Kolen et al, 2023) to electroneutral H+-sialic acid symport by sialin (Morin et al, 2004; Hu et al, 2023). Although this protein family has been studied for several decades, the mechanistic basis of SLC17 organic anion transport remains insufficiently understood.

The bacterial homolog DgoT (from *E. coli*) transports negatively charged galactonate in symport with more than one proton and, thus, differs from mammalian SLC17s in both its main substrate and transport stoichiometry (Leano et al, 2019). It belongs to the large major facilitator superfamily (MFS). MFS family members exhibit a distinct topology that includes 12 transmembrane (TM) helices, with the substrate-binding pocket located between two pseudo-symmetrical 6-TM bundles in the N-domain (TM1–TM6) and C-domain (TM7–TM12) (Quistgaard et al, 2016; Drew et al, 2021). DgoT has been crystallized in two conformations (Leano et al, 2019): (i) inward-facing and (ii) outward-facing with galactonate bound to the central binding site (Fig. 1). These structures revealed four charged residues within the transmembrane domains, D46, R47, R126, and E133, with the last three conserved across the SLC17 family. The high structural similarity with vesicular glutamate transporters (VGLUTs) (Li et al, 2020) and lysosomal sialic acid transporter (Hu et al, 2023) (Fig. EV1; Appendix Fig. S1), together with differences in substrate selectivity and transport stoichiometry, makes DgoT an attractive model to study the atomistic basis of coupled transport in the SLC17 family. Here, we present a comprehensive study of DgoT transport using a combination of in vitro experiments and molecular dynamics (MD) simulations.

## Results

### DgoT mediates the coupled symport of two H+ and one D-galactonate

We studied DgoT transport using solid-supported membrane-based electrophysiology (SSME) with purified DgoT reconstituted into proteoliposomes. In such recordings, transport activity is induced by fast solution exchange. The resulting changes in proteoliposomal membrane potential are converted into an

[1]Institute of Biological Information Processing (IBI-1), Molekular- und Zellphysiologie, Forschungszentrum Jülich, 52425 Jülich, Germany. [2]Institute for Advanced Simulation (IAS-5) and Institute of Neuroscience and Medicine (INM-9), Computational Biomedicine, Forschungszentrum Jülich, 52425 Jülich, Germany. [3]JARA-HPC, Forschungszentrum Jülich, 54245 Jülich, Germany. [4]Department of Physics, RWTH Aachen University, 52056 Aachen, Germany. [5]JARA Institute Molecular Neuroscience and Neuroimaging (INM-11), Forschungszentrum Jülich, 52425 Jülich, Germany. [6]Cécile and Oskar Vogt Institute for Brain Research, University Hospital Düsseldorf, Medical Faculty, Heinrich Heine University Düsseldorf, 40225 Düsseldorf, Germany. [7]Present address: Department of Biochemistry and Biophysics and Science for Life Laboratory, Stockholm University, Stockholm, Sweden. [8]These authors contributed equally: Mercedes Alfonso-Prieto, Christoph Fahlke. ✉E-mail: c.fahlke@fz-juelich.de

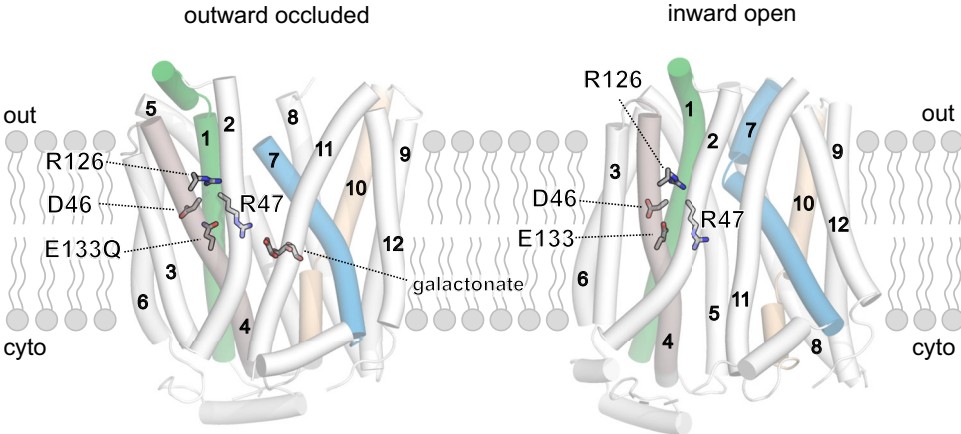

**Figure 1.  X-ray crystal structures of E133Q (*left*) and WT (*right*) DgoT.**

The positions of the four charged residues in transmembrane helices and of the bound galactonate are indicated.

electrical signal and detected by a measuring electrode via capacitive coupling (Schulz et al, 2008; Bazzone et al, 2021).

Figure 2A depicts SSME currents after galactonate concentration jumps at three different pHs. At neutral and alkaline pH, the time course for current decay decreased at higher lipid–protein ratios (LPRs; illustrated for pH 8.0 in the inset). This result indicates that the measured currents are caused by electrogenic DgoT transport (Bazzone et al, 2021). Transport currents assumed a maximum amplitude at pH 7.5 (blue line) and decreased at higher pH values (Fig. 2A; pH 9.0, black line). Under asymmetrical ionic conditions (i.e., with an alkaline pH outside, but not inside the liposomes), the peak current was reduced (Fig. 2B, bottom). This suggests that reduced transport at alkaline pH is caused by proton depletion at the binding site outside the liposomes. Under symmetrical pH conditions, the current decay was biphasic at pH values lower than 7.5 (Fig. EV2): the fast component represents conformational changes triggered by galactonate binding and the slow component is associated with the transport activity of the protein. At pH values lower than 6, only the pre-steady state, but not the transport component, was observed (Fig. 2A, red line). Acidic pH inside, but not outside, the liposomes inhibits transport activity, as demonstrated under asymmetrical ionic conditions (Fig. 2B, top). We conclude that current reduction under conditions of acidic or alkaline pH is regulated by two different processes: proton binding from the outside and proton release to the inside of the proteoliposomes, respectively. Thus, analysis of the peak currents allowed the determination of apparent $pK_a$ values for proton binding and release (Fig. 2C).

The transport currents generated by DgoT were positive (Figs. 2A and EV2), indicating that at least two protons are transported with each galactonate molecule, as previously reported (Leano et al, 2019). We used a reversal potential assay to determine the transport stoichiometry (Thomas et al, 2021). In SSME, no voltage is applied to the membrane and concentration gradients are the only driving force for coupled transport. The free energy for the coupled transport reaction is given by:

$$\Delta G = n\Delta\mu_i + m\Delta\mu_s,$$

where $n$ and $m$ are number of protons and galactonate molecules transported together. The current reverses at $\Delta G = 0$:

$$\frac{n}{m} = -\frac{\Delta\mu_s}{\Delta\mu_i} = -\frac{RT\,ln\left(\frac{[galactonate]_{in}}{[galactonate]_{out}}\right)}{RT\,ln\left(\frac{[H^+]_{in}}{[H^+]_{out}}\right)} = \frac{ln\left(\frac{[galactonate]_{in}}{[galactonate]_{out}}\right)}{ln\left(\frac{[H^+]_{out}}{[H^+]_{in}}\right)}$$

Thus, plotting the transported charge against the ratio of the electrochemical gradients enables the transport stoichiometry to be determined (Thomas et al, 2021).

We preloaded proteoliposomes with galactonate-containing internal solutions and applied sets of external solutions to generate an outwardly directed proton concentration gradient at various galactonate concentration gradients. The currents obtained with empty liposomes—used as the negative control (Appendix Fig. S2)—were subtracted. Null transport occurred at the gradient ratios corresponding to a stoichiometry of 2 $H^+$:1 galactonate (Fig. 2D,E).

To investigate the binding order of $H^+$ and galactonate, we measured changes to the external pH of solutions containing purified detergent-solubilized protein upon the addition of galactonate (Soskine et al, 2004). If galactonate binding precedes proton association, we would expect the pH to increase upon galactonate addition. However, addition of 10 mM galactonate to purified DgoT in unbuffered detergent-containing solutions did not elicit any change in pH (Fig. 2F). This result suggests that the substrate binds to the protonated transporter and does not induce protonation of DgoT. Therefore, we conclude that DgoT binds two $H^+$ prior to galactonate association.

## Protonation of key residues regulates extracellular gate dynamics

We studied $H^+$ and galactonate binding to outward-facing DgoT using unbiased all-atom MD simulations. For this, we used the crystal structure of E133Q DgoT (Leano et al, 2019) (PDB ID: 6E9O) as the starting conformation after reverting the mutation

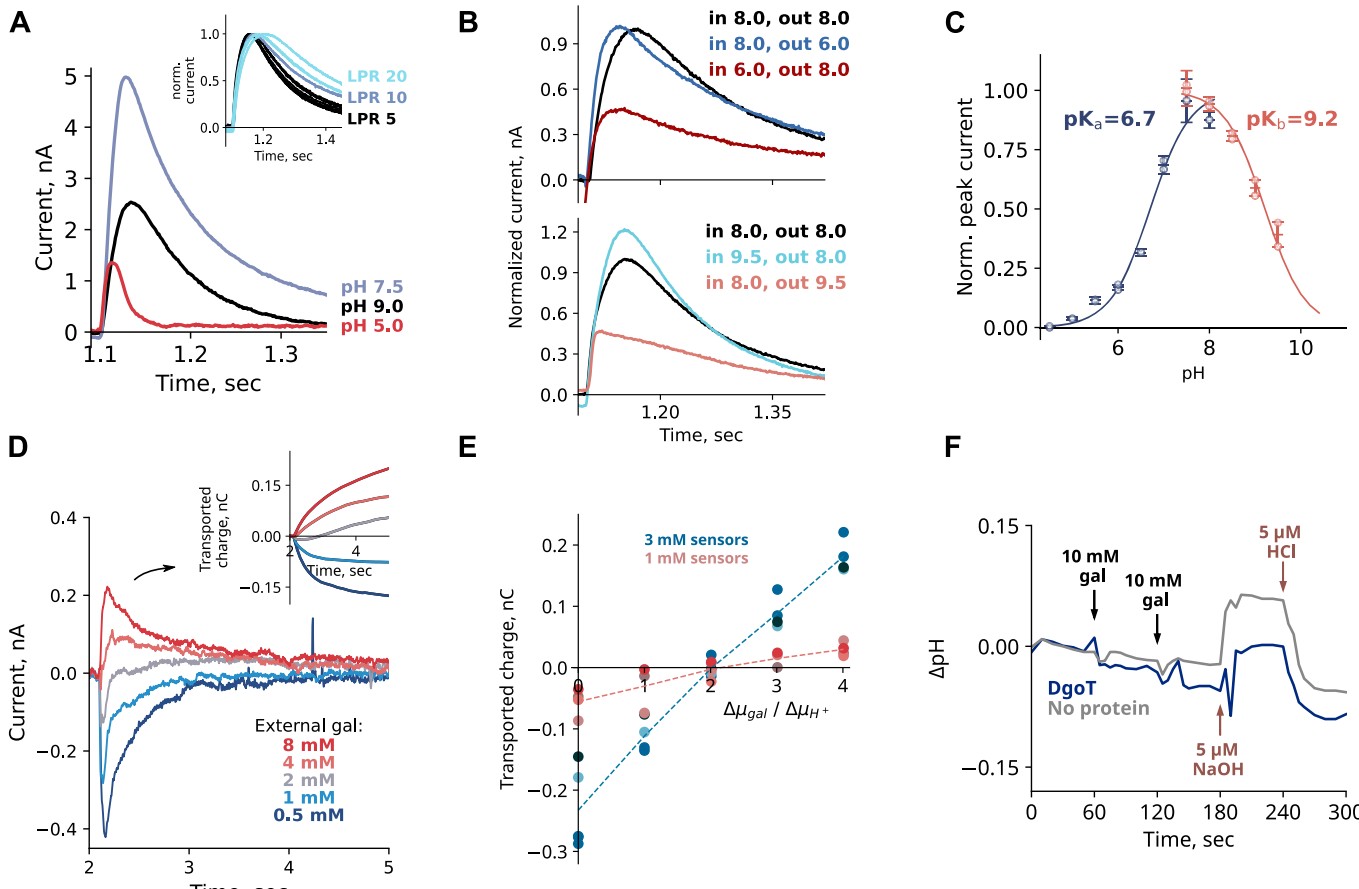

**Figure 2. Functional characterization of DgoT.**

(A) pH dependence of WT DgoT peak currents measured by SSME upon application of 10 mM D-galactonate concentration jump. The inset shows transport currents at pH 8.0 from liposomes reconstituted with different lipid-to-protein ratios (LPR). Currents were normalized to their peak value for comparison of the current decay. (B) Transient currents under symmetrical (black line) or asymmetrical conditions. Peak current decreases if acidic pH is applied inside liposomes (top) or alkaline pH is applied outside (bottom). (C) Peak currents and determination of apparent $pK_a$ values. The solid lines are fits to the data using the equations described in Methods. Experiments were performed in duplicate on two independent sensors. The error bars represent standard deviations. (D) Representative transport current traces used for the determination of the transport stoichiometry. The inset shows time dependence of transported charge obtained by integration of current traces. (E) Plots of transported charge *versus* ratio of the galactonate and proton chemical potentials. (F) Changes in pH after addition of substrate to purified DgoT in detergent. Additions of 10 mM of galactonate are indicated by the black arrows. 5 µM HCl or NaOH were added to induce a pH shift comparable with the expected changes in $H^+$ concentration due to proton binding to DgoT. Source data are available online for this figure.

(see "Methods"). First, we investigate the effect of protonation on *apo* protein dynamics. The crystallographic galactonate molecule was removed from the binding site, and four different systems were generated with D46 and E133 individually protonated or deprotonated. We observed reproducible movements of the TM1 and TM7 transmembrane helices, with most flexibility in the most extracellular segments (around residues 48–52 of TM1 and 271–275 of TM7). This behavior resembles the proposed role of TM1 and TM7 as gating helices that mediate transitions between open and occluded conformations, as in other MFS members (Smirnova et al, 2014; Qureshi et al, 2020; Feng et al, 2021).

When D46 and E133 were unprotonated, the extracellular gate (defined as distance between the center of mass of the Cα atoms of residues 48–52 and residues 271–275) was mostly closed (Fig. 3A,D). Protonation of both D46 and E133 locked the gate in an open conformation (Fig. 3B,D), and protonation of only one

of these residues only partially opened the gate (Fig. 3D; Appendix Fig. S3A). These changes in position of the gating helices were coupled to local rearrangements in the N-terminal domain. Charged, but not protonated D46 and E133 interact with R126 and R47, respectively. After protonation, the side chain of D46 moves away, allowing R126 to interact with N49. This results in rotation of TM1 and stabilization of the open state of the extracellular gate (compare Fig. 3A,B). In addition, protonation of E133 releases R47, which then becomes available to interact with the carboxyl group of galactonate.

To probe possible pathways by which D46 and E133 can be both protonated in *apo* DgoT, we analyzed the water accessibility of the two carboxyl groups from the extracellular solution (Fig. EV3). In trajectories with D46 and E133 deprotonated and with the extracellular gate partially open, both carboxyl groups were accessible from the bulk (Fig. EV3A), allowing the simultaneous protonation of both sites. Alternatively, the two protonation events

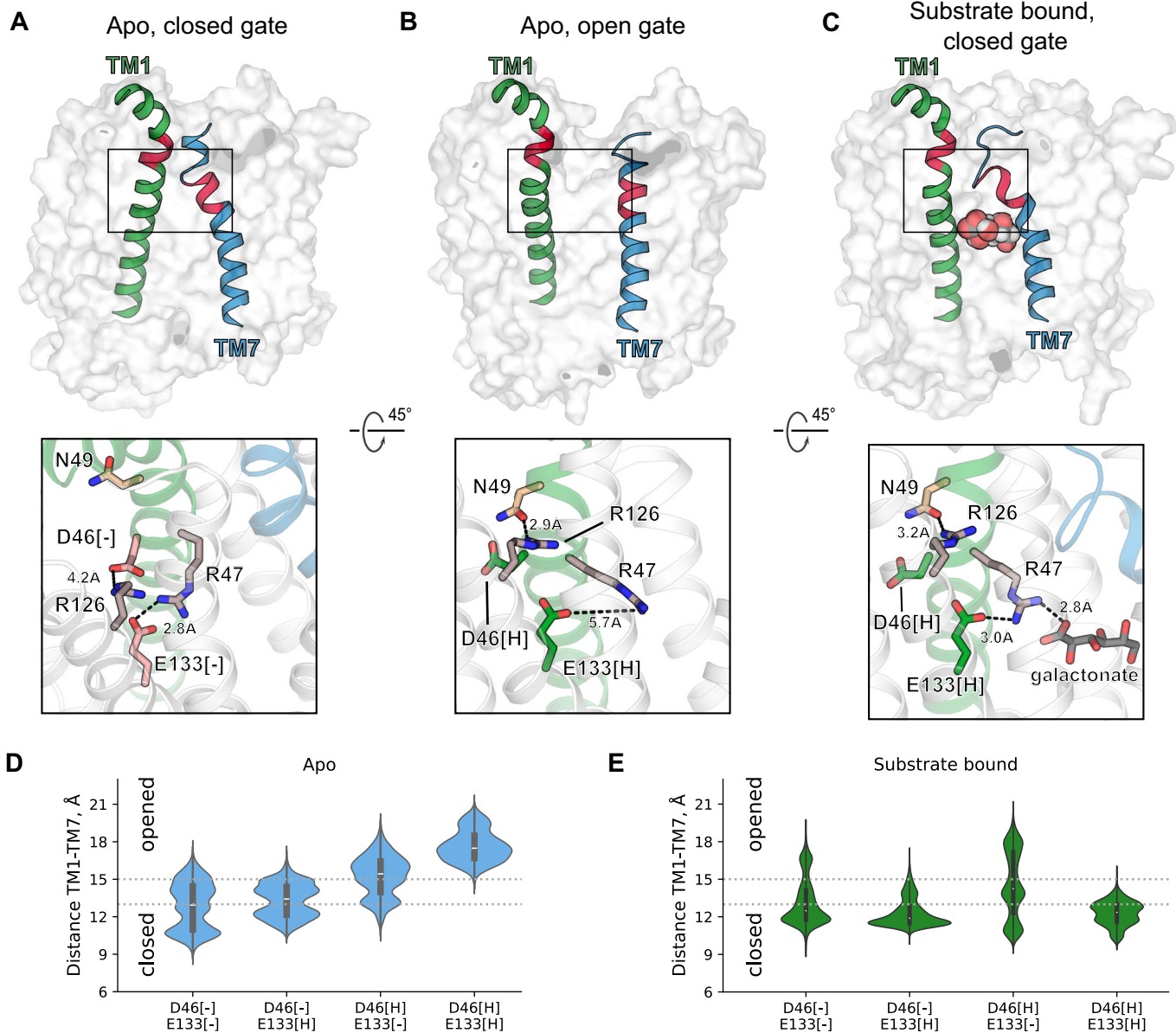

**Figure 3. Unbiased MD simulations reveal protonation- and substrate-dependent dynamics of extracellular gate.**

(A–C) Top row: snapshots showing the arrangement of the gating helices TM1 and TM7 in unbiased MD simulations with DgoT in outward-facing conformation with D46 and E133 deprotonated (**A**), protonated (**B**) and galactonate bound (**C**). The red colored parts of the helices correspond to residues 48–52 (TM1) and residues 271–275 (TM7). Bottom row: representative snapshots showing interactions around D46 and E133. (**D**, **E**) Probability densities for extracellular gate opening in *apo* (**D**) and galactonate-bound (**E**) MD simulations with DgoT in the outward-facing conformation with different protonation states of D46 and E133 (for *apo* simulations: $n = 8$ for D46 and E133 deprotonated, $n = 4$ for the rest; for galactonate-bound simulations: $n = 5$ for D46 and E133 deprotonated and $n = 4$ for the rest). The TM1–TM7 distance is measured as distance between center of mass of Cα atoms of residues 48–52 and residues 271–275. Horizontal lines indicate closed (<13 Å) and open (>15 Å) states of the extracellular gate. The central white dots represent the medians, thick black lines represent ranges between the 25th percentile (Q1) and the 75th percentile (Q3), the top and bottom points of the violin extend to the minimum and maximum values of the kernel density estimate (KDE). Source data are available online for this figure.

can be consecutive. After initial protonation of D46, the carboxyl group of E133 is water accessible even in trajectories with the narrowest distance between TM1 and TM7 and can therefore be protonated from the solution (Fig. EV3B). We also compared the electrostatic potential in the vicinity of the carboxyl groups (calculated using *g_elpot* (Kostritskii et al, 2021)). The potential near E133 was negative, although higher in the system with D46 protonated (Fig. EV3C), suggesting that glutamate can accept the

proton from the extracellular solution after aspartate. If instead E133 is protonated first, subsequent protonation of D46 from the extracellular solution would be less probable, as water molecules near its carboxyl group are separated from the bulk (Fig. EV3D). However, the carboxyl groups of deprotonated D46 and protonated E133 are often within ~3 Å distance from one another (Fig. EV3E). Thus, we analyzed the possibility of proton transfer from E133 to D46 in this system by quantum mechanics/molecular mechanics

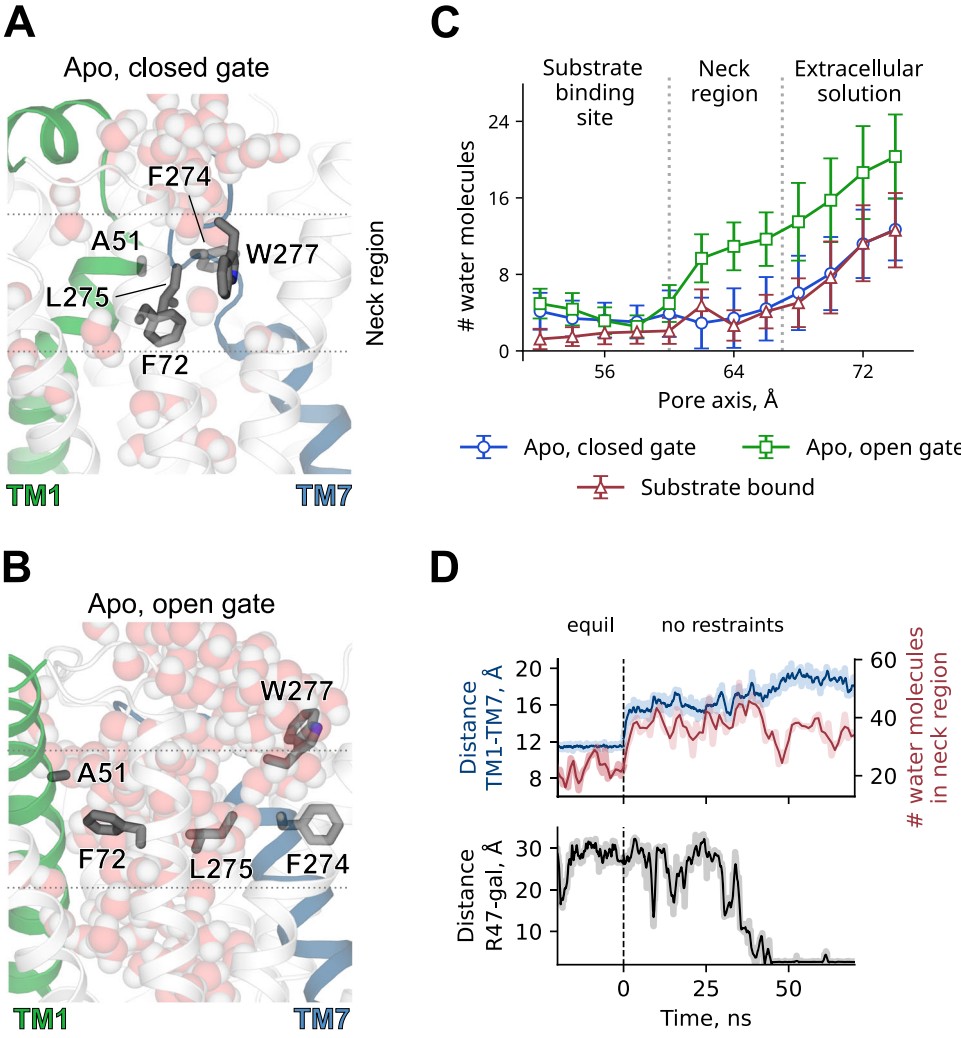

**Figure 4. Hydrophobic interactions at the level of the extracellular gate.**

(A, B) Snapshots showing water molecules distribution in simulations with DgoT in outward-facing conformation with closed (A) and open (B) extracellular gate. Representative snapshots were taken from unbiased MD simulations with both D46 and E133 either deprotonated (A) or protonated (B). (C) Hydration profile of the pore represented by the number of water molecules in 2 Å sections along the pore axis. Trajectories with following parameters were used for analysis: *apo*, closed gate—D46 and E133 deprotonated ($n = 8$); *apo*, open gate and substrate bound—D46 and E133 protonated ($n = 4$ for both). In *apo* simulations only frames with minimum distance between side chains of F72 and W277 <12.5 Å (blue line) or >12.5 Å (green line) were used. The error bars represent standard deviation. (D) Time course of extracellular gate opening (measured as TM1–TM7 distance, as in Fig. 3), number of water molecules in 10 Å section near extracellular gate (z coordinate between 58 and 68 Å) and galactonate binding to DgoT (measured as minimum distance between galactonate molecule and guanidinium group of R47) in a trajectory with both D46 and E133 protonated. Shaded lines represent raw data from the trajectory, solid lines are moving averages. Source data are available online for this figure.

(QM/MM) thermodynamic integration, which yielded an energy barrier of only ~4 kcal/mol (Fig. EV3F). Thus, D46 can accept the proton from E133 and the latter can subsequently be protonated from the bulk, as discussed above (Fig. EV3B). We thus conclude that D46 and E133 can be both protonated, either simultaneously or consecutively, from the extracellular solution.

We next performed unbiased simulations with the *apo* outward-facing structure in presence of 100 mM galactonate in solution. Since galactonate is predicted to have a pK$_a$ of 3.39 in bulk water, galactonate likely binds to DgoT in its deprotonated form. Before starting the MD simulations, D46 and E133 were protonated or deprotonated individually. For each system, at least one of the two key residues was protonated and five replicates were used. We

observed spontaneous galactonate-binding events in two simulations in which both D46 and E133 were protonated and in one simulation in which only E133 was protonated. In all cases, opening of the extracellular gate preceded the entry of galactonate into the binding site (Fig. 4D; Appendix Fig. S4).

## Substrate binding induces closure of the extracellular gate

To further investigate the structural changes resulting from substrate binding, we ran unbiased simulations of outward-facing DgoT with bound galactonate (see "Methods"). These simulations revealed a reduced flexibility of gating helices TM1 and TM7 (Fig. 3C) and the

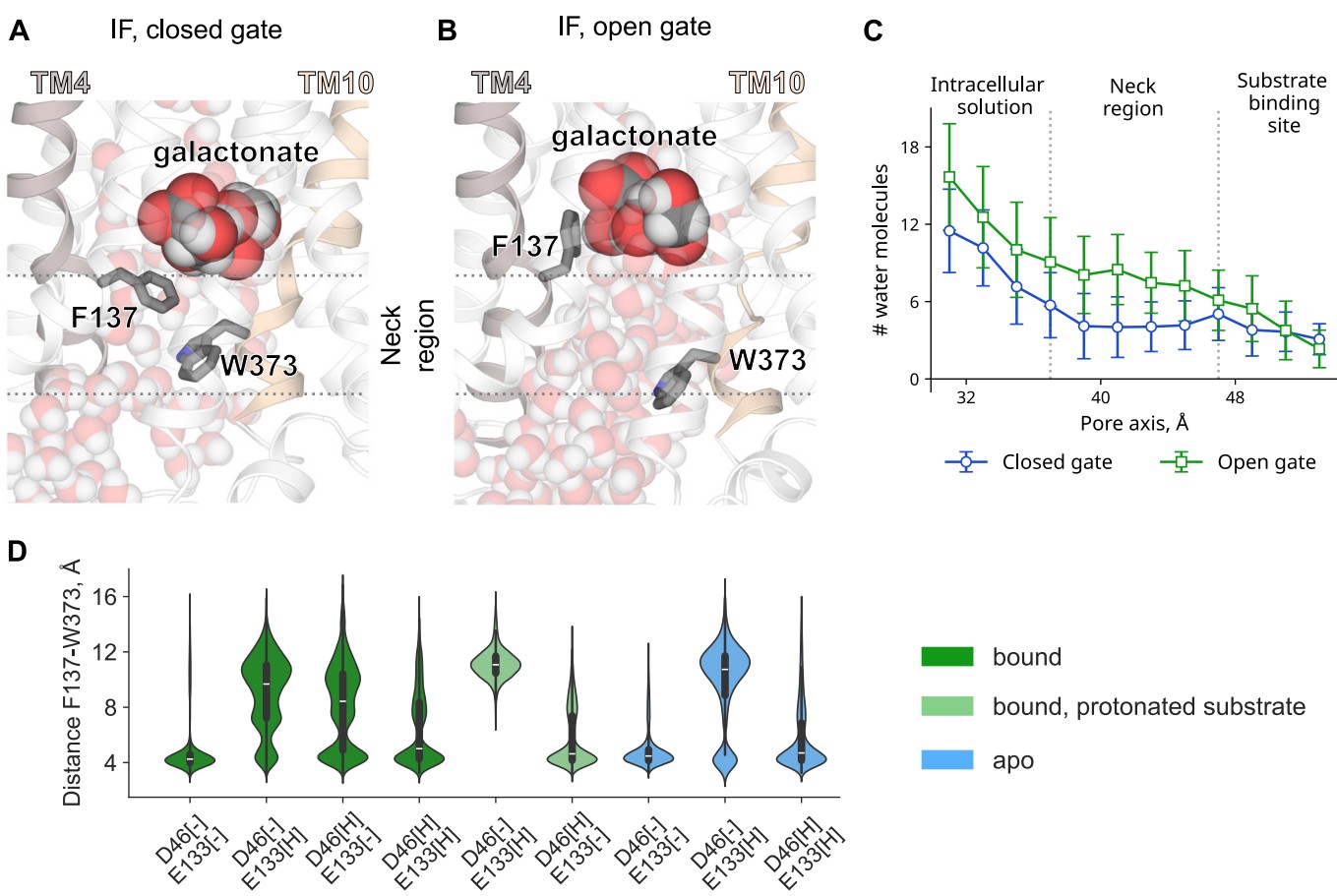

**Figure 5. A hydrophobic lock within the intracellular gate regulates access to the binding site.**

(A, B) Snapshots showing water molecule distribution in simulations with DgoT in inward-facing conformation with closed (A) and open (B) intracellular gate. Representative snapshots were taken from unbiased MD simulations with D46 and E133 protonated (A) or only E133 protonated (B). (C) Hydration profile of the pore, represented by the number of water molecules in 2 Å sections along the pore axis. Trajectories with following parameters were used for analysis: closed gate—D46 and E133 protonated ($n = 4$); open gate—D46 deprotonated, E133 protonated ($n = 10$). Substrate was bound to the protein in both cases. Only frames with minimum distance between side chains of F137 and W373 < 8 Å (blue line) or >8 Å (green line) were used. The error bars represent standard deviation. (D) Probability densities for intracellular gate opening (measured as distance between side chains of F137 and W373) in MD simulations with DgoT in inward-facing conformation with different protonation states of D46 and E133 and galactonate and various substrate occupancies (for simulations with deprotonated galactonate bound: $n = 10$ for simulations with only D46 or only E133 protonated, $n = 5$ for simulations with both D46 and E133 protonated; for the rest $n = 4$). The central white dots represent the medians, thick black lines represent ranges between the 25th percentile (Q1) and the 75th percentile (Q3), the top and bottom points of the violin extend to the minimum and maximum values of the kernel density estimate (KDE). Source data are available online for this figure.

extracellular gate predominantly in a closed conformation (Fig. 3E; Appendix Fig. S5), preventing galactonate from being released back to the extracellular solution. Although the gate could close regardless of the protonation state of key residues in substrate-bound DgoT, the equilibrium was shifted toward the conformation with closed extracellular gate in simulations with only E133 protonated or with both D46 and E133 protonated (Fig. 3E).

Protonation of E133 permits the R47 side chain to simultaneously interact with the carboxyl groups of galactonate and E133, resulting in a more compact arrangement of the charged residues (Fig. 3C). Substrate interaction with residues on both gating helices facilitates bending of the TM7 extracellular segment toward TM1. Hydrophobic interactions also play a role in forming occluded conformations. In the outward-facing occluded state, substrate-binding site is separated from the extracellular solution by hydrophobic residues located in TM1, TM2, and TM7 (F72, F274, and W277, respectively; Fig. 4A). When the extracellular gate opens, the position of the side chains of the three hydrophobic residues

changes to allow more water molecules to flow between TM1 and TM7 (Fig. 4B) and galactonate to enter the binding site. Once the substrate is bound and the protein adopts an occluded conformation, the number of water molecules is reduced (Fig. 4C). Water entry appears to be exclusively determined by the extracellular gate opening and not by the charge of D46 and E133 (Appendix Fig. S5). This resembles a "clamp-and-switch" mechanism (Quistgaard et al, 2016), in which occlusion of the binding site precedes a rocker-switch-type rotation on the N- and C-terminal domains, causing the binding site to become exposed to the opposite side (Qureshi et al, 2020).

## Structural basis of substrate release

We next ran unbiased simulations with the inward-facing structure of DgoT after placing a galactonate molecule into the binding site. The position for the substrate molecule was determined by aligning the crystal structures of the *apo* protein in the inward-facing state (PDB ID:

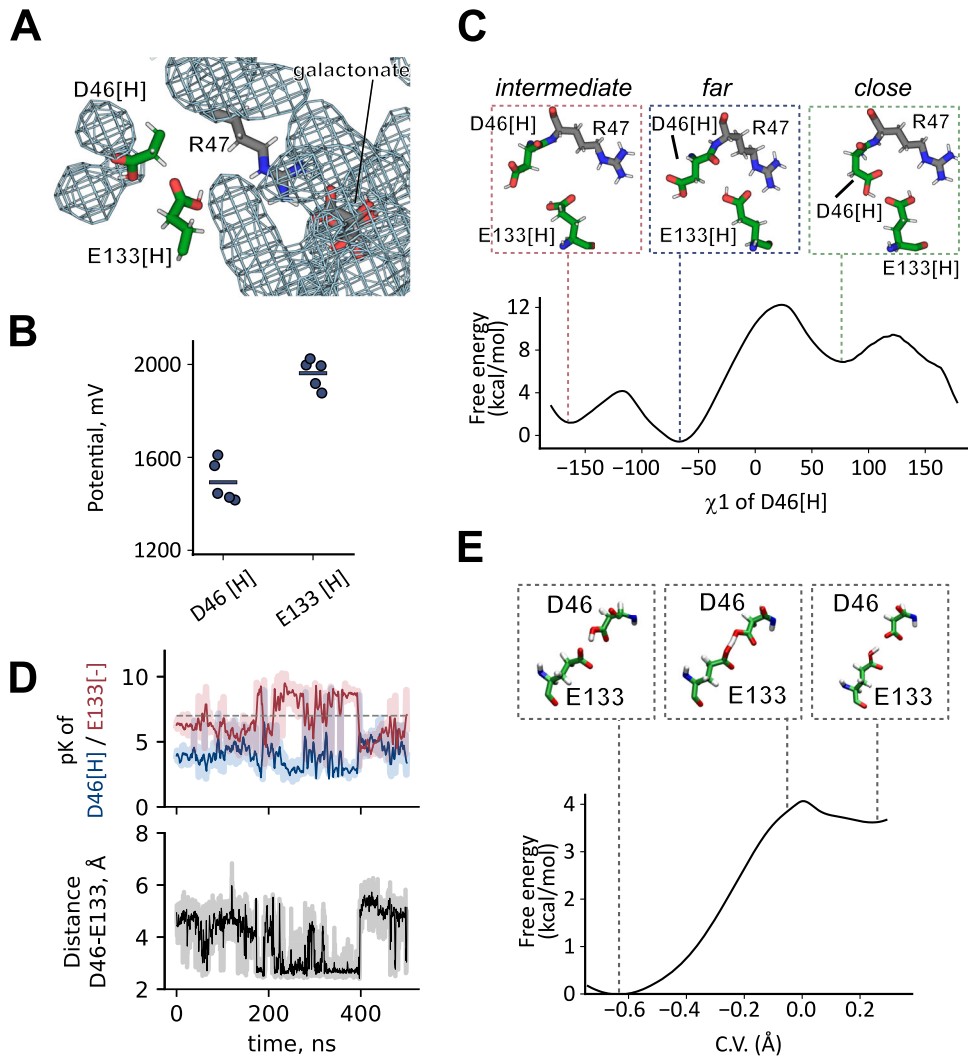

**Figure 6. Proton release from D46.**

(**A**) Water occupancy map in unbiased simulations with D46 and E133 protonated, contoured at an occupancy level of 0.1. (**B**) Electrostatic potential at the carboxyl groups of protonated D46 and E133 in the corresponding unbiased MD replica simulations ($n = 5$). (**C**) Classical well-tempered metadynamics simulations of the conformational change of protonated D46. (**D**) Time course of pKa of D46 and E133 and distance between side chains of D46 and E133 in one of simulations with inward-facing DgoT with D46 protonated, E133 deprotonated and $\chi1$ of D46 restrained at $+70°$ (close-D46). Galactonate was present in the binding site in its deprotonated (**D**) or its protonated (**E**) form. Shaded lines represent raw data from the trajectory, solid lines are moving averages. (**E**) Free energy profile for the proton transfer between D46 and E133, computed at the QM (BLYP)/MM level. The insets show representative starting, transition state, and final configurations. Error bars are omitted since they are smaller than the marker size. Source data are available online for this figure.

6E9N) with the galactonate-bound outward-facing state (PDB ID: 6E9O). Galactonate likely binds to outward-facing DgoT in its deprotonated form, but we also considered the possibility that galactonate could be protonated in the binding pocket due to changes in the local environment and then released in its protonated form (see "Methods"). In 7 out of 38 individual 500 ns unbiased MD simulations, we observed spontaneous galactonate release (Appendix Table S1; Appendix Fig. S6). In all cases, the substrate passed between TM4 and TM10, which act as gating helices on the intracellular side of the protein. The intracellular segments of these helices (around residues 139–143 and 372–375, respectively) showed a high degree of flexibility in these simulations. The side chains of F137 and W373 formed a hydrophobic lock similar to the one at the extracellular gate (Fig. 5A),

and galactonate could only pass through this neck region if a water-filled pore was formed between the side chains of F137 and W373 (Fig. 5B,C).

All spontaneous release events were observed in systems containing deprotonated D46, which suggests that the proton must be released from this residue prior to galactonate unbinding. Figure 5D shows probability densities for intracellular gate opening (measured as the minimum distance between the side chains of F137 and W373) for various protonation and substrate-binding states (see also Appendix Fig. S7). The intracellular gate mostly assumed a closed conformation in the substrate-bound double protonated system. Deprotonation of D46 favored the open state of the gate in systems with either protonated or deprotonated

substrate. When galactonate is protonated, the intracellular gate has limited flexibility and becomes locked in the open state, when only E133 is protonated, and in the closed state, when only D46 is protonated. After release of the substrate and both protons, the intracellular gate closes, which will permit reorientation of the *apo* transporter to the outward conformation. We conclude that D46 deprotonation increases the probability of galactonate release from the inward-facing DgoT by promoting the opening of the intracellular gate.

## Proton release from inward-facing DgoT

Galactonate only spontaneously dissociated from inward-facing DgoT, when D46 was deprotonated (Appendix Table S1). However, in MD simulations of DgoT in which both D46 and E133 were protonated, the D46 carboxyl group was sequestered (Fig. 6A), making direct deprotonation to the intracellular solution unlikely. In addition, the electrostatic potential (calculated using *g_elpot* (Kostritskii et al, 2021; Kostritskii and Machtens, 2023)) was higher on the E133 carboxyl group than on D46 (Fig. 6B), indicating that E133 is likely to be the first residue to release its proton. PROPKA (Olsson et al, 2011) predicted $pK_a$ values of D46 mostly above 8, while $pK_a$s of E133 were often below 7 (Fig. EV4A). These observations suggest that D46 donates its proton to E133 after the latter had released its proton. To further probe this hypothesis, we used PROPKA to estimate $pK_a$ values in simulations with only D46 protonated. In unbiased classical MD trajectories, in which D46 and E133 side chains were far from each other (Fig. EV4B), D46 was predicted to have elevated $pK_a$ and E133 to have a low $pK_a$ (Fig. EV4C). We then explored a different configuration in which the carboxyl groups of D46 and E133 are interacting, by applying a harmonic potential on the $\chi_1$ (N-C$\alpha$-C$\beta$-C$\gamma$) dihedral angle of D46. The restraint at $\chi_1 \sim +70°$ allowed sampling the formation of a hydrogen bond between the carboxyl groups of D46 and E133, which was associated with a swap of $pK_a$ values of the two acidic residues (Figs. 6D and EV4D,E). This suggests that the proximity of the two carboxyl groups is a necessary prerequisite for proton transfer from D46 to E133.

Thus, we next calculated the free energy associated with conformational change as a function of the N-C$\alpha$-C$\beta$-C$\gamma$ ($\chi_1$) dihedral angle of D46 using well-tempered metadynamics (WT-MTD) (Barducci et al, 2008; Dama et al, 2014). Figure 6C shows the presence of a minimum at $\chi_1 = -70°$ (*far*-D46), corresponding to the conformation sampled in unbiased MD simulations. However, after passing through an *intermediate* conformation ($\chi_1 = -170°$) and crossing over a barrier of ~10 kcal mol$^{-1}$, another minimum is explored (hereafter denoted as *close*-D46, characterized by $\chi_1 = +70°$), which has a higher free energy (by ~8 kcal mol$^{-1}$). In this conformation, the carboxyl group of D46 is H-bonded to E133, i.e. in a proton transfer competent conformation.

E133 may release its proton either to the solvent or to galactonate (Fig. 6A). Hydrated excess protons can reorganize the surrounding water wires and facilitate further proton translocation (Peng et al, 2015; Li and Voth, 2021a). Therefore, we first studied proton transfer from E133 to the adjacent water molecule to form a hydronium ion. QM/MM thermodynamics integration (TI) calculations showed that the free energy barrier for proton release from E133 is ~8 kcal mol$^{-1}$ (Appendix Fig. S8). Although the accuracy of

the calculated free energy profile is limited due to the use of a simple difference of distances constraint (Sprik, 2000; Li and Voth, 2021b), modeling the ion pair between the negatively charged E133 and the adjacent hydronium ion allowed us to explore the reorganization of the surrounding water molecules that connect the excess proton with both galactonate and the solution. To represent the diverse range of possible H-bond networks, we selected seven structures (see Methods) for use in unbiased QM/MM MD simulations, in which the quantum part was treated at the density functional theory (DFT)-BLYP level (Lee et al, 1988; Becke, 1988). In simulations with three of the selected structures, the galactonate carboxyl group accepted the proton after a few picoseconds, either via water molecules only or via water molecules along with the substrate hydroxyl group (Appendix Fig. S9; Appendix Table S2). In simulations with three other structures, the proton was transferred to one of the molecules of the water wire, without substrate protonation. In simulations with the seventh structure, the proton was shared between the substrate and a neighboring water molecule. Therefore, despite the limited statistics and underestimation of the proton transfer barriers due to the BLYP functional used (Mangiatordi et al, 2012), our QM/MM simulations suggest that, upon D46 rotation and deprotonation of E133, proton transfer can occur either towards galactonate or to the solvent. The released proton might eventually leave the protein either bound to the substrate or through the solvent via a Grotthuss mechanism.

Finally, we performed QM/MM TI calculations to study the proton transfer from *close*-D46 to negatively charged E133, yielding a free energy barrier of around 4 kcal mol$^{-1}$ (Fig. 5D). Altogether, the stepwise mechanism for D46 deprotonation studied here (Appendix Fig. S10) involves first the D46 side chain rotation, followed by proton transfer from E133 to the solvent and/or galactonate and subsequent proton transfer from D46 to E133. The results of our classical and QM/MM enhanced sampling simulations indicate that such mechanism is feasible. However, the limited accuracy of the energetics calculated on the basis of 1D free energy profiles (Fig. 6C,E; Appendix Fig. S8) precludes the estimation of the overall timescale for proton release from D46 in the inward facing state. Moreover, there might exist other proton transfer pathways not investigated here, as shown for other transporters (Parker et al, 2017; Liu et al, 2021; Swanson, 2022; Li et al, 2022; Liu et al, 2024).

## Free energy landscapes suggest that reorientation of the empty transporter is a rate-limiting step of the cycle

Both inward- and outward-facing DgoT conformations display some flexibility; however, we did not observe transitions between these two states in single trajectories. Since a quantitative description requires multiple observations of these slow processes, we used Markov state modeling (Husic and Pande, 2018) to stitch together short simulation data and analyze the transition probability between all conformational states. We featurized the trajectory data using a set of interdomain C$\alpha$ atomic distances and applied time-lagged independent component analysis (tICA) to find the slowest components in the dataset. The first tICA-eigenvector (tIC1) discriminates between the inward-and outward-facing conformations (Fig. 7A), and the second (tIC2) correlates with the (open or closed) state of the

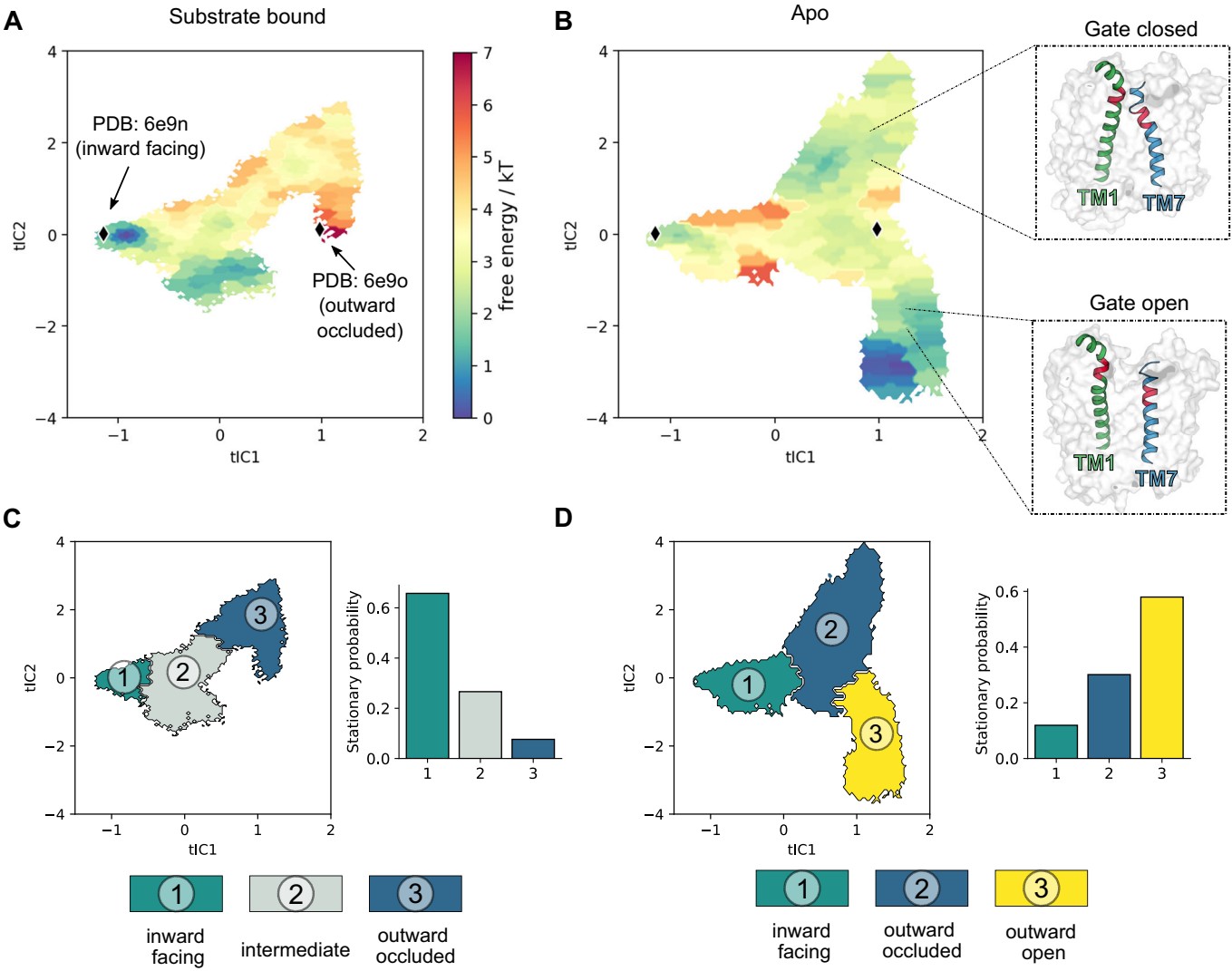

**Figure 7. Energetical description of major conformational changes.**

(A, B) Free energy landscape for DgoT in substrate-bound (A) or *apo* (B) state. Protein conformations captured in DgoT crystal structures were projected onto the time-lagged independent component analysis (tICA) space (black points). Representative snapshots illustrate how the value of tIC2 correlates with the degree of opening of the extracellular gate. (C) Left: coarse representation of intermediate metastable states obtained with Perron-cluster cluster analysis (PCCA) for substrate-bound DgoT. Right: Stationary probabilities of such metastables states. (D) Same representation as in (C), for *apo* DgoT. Source data are available online for this figure.

extracellular gate (Fig. 7B; Appendix Fig. S11). Two systems were chosen as most relevant for transport: (i) galactonate-bound DgoT with protonated D46 and E133 (responsible for translocation of the substrates across the membrane) and (ii) *apo* DgoT with deprotonated D46 and E133 (responsible for reorientation of the empty transporter after substrate release). For both systems, we sampled all relevant intermediate conformations, with a total of ~26 μs (substrate-bound) and ~35 μs (*apo*) unbiased MD simulations, and constructed Markov state models (Fig. 7A,B). The free energy surfaces for both systems revealed a high energy barrier separating the inward- and outward-facing conformations. In the presence of galactonate, the barrier was lowered and the inward-facing state was the most energetically favorable. In contrast, the system with the *apo* protein preferred outward-facing conformations such that DgoT favors inward galactonate

transport. Moreover, *apo* DgoT easily switches between outward-occluded and outward-open states, whereas substrate-bound DgoT adopts only occluded conformations. Perron-cluster cluster analysis (PCCA) (Deuflhard and Weber, 2005), used to identify metastable states, revealed a difference in the direction of conformational changes between the *apo* and galactonate-bound states (Fig. 7C,D), as previously observed in the free energy surfaces (Fig. 7A,B). For substrate-bound DgoT, metastable states correspond to inward-facing, intermediate occluded, and outward-occluded states, with the highest probability for the inward conformation (Fig. 7C). For *apo* DgoT, PCCA identified inward-facing, outward-occluded, and outward-open states (lower to higher probability). The outward-open state was not adopted by substrate-bound DgoT: it was only adopted by *apo* DgoT, for which it represents the state with highest stationary

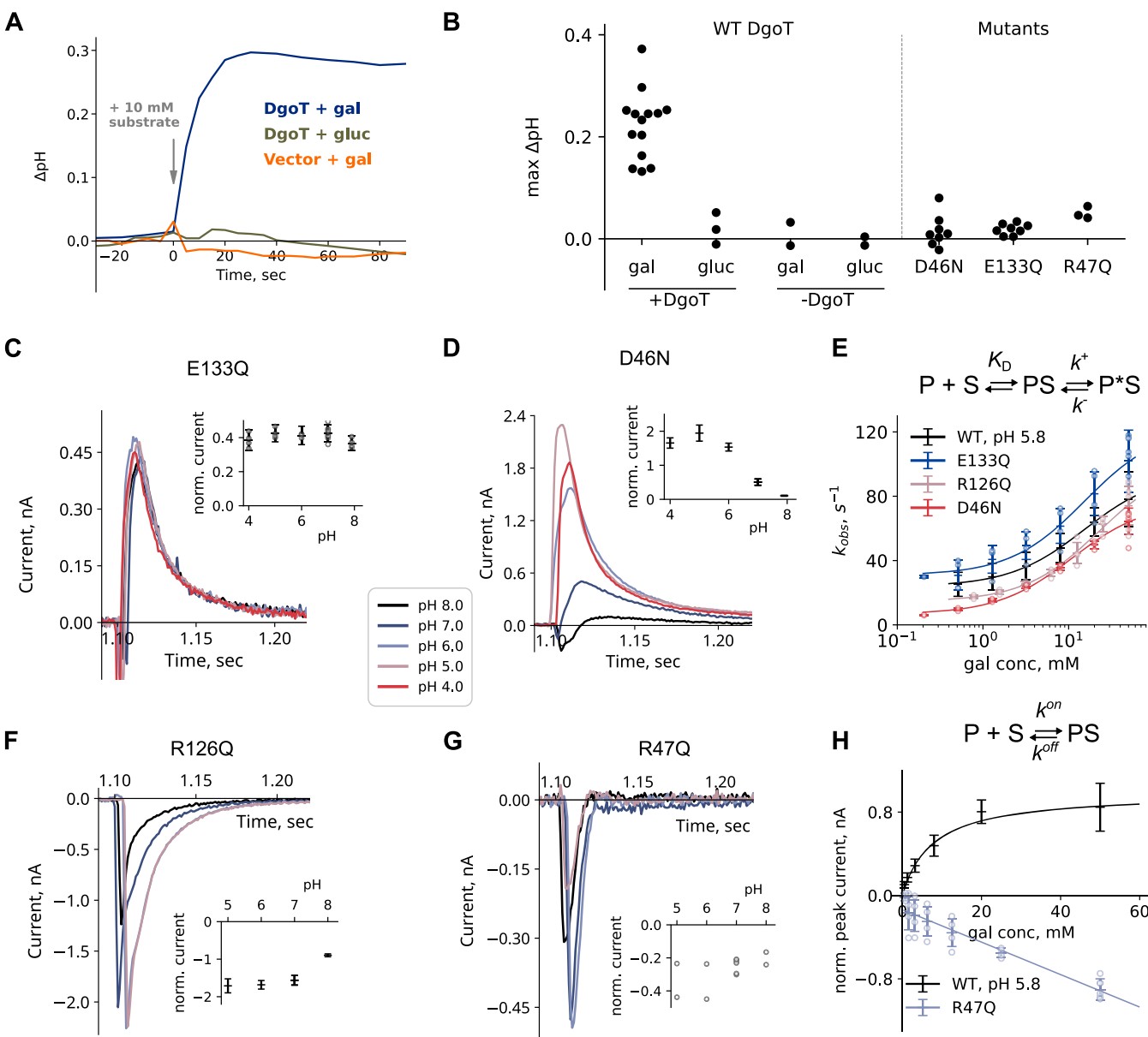

**Figure 8. Neutralization of transmembrane charged residues abolish DgoT galactonate transport.**

(A) pH changes induced by addition of galactonate, or gluconate to *E. coli* cells either expressing WT DgoT or transformed with the same vector without the DgoT gene. (B) Maximum pH changes observed in experiments with *E. coli* cells expressing different DgoT variants (*n* = 14 for WT with galactonate added, *n* = 3 for WT with gluconate added, *n* = 2 for vector without DgoT, *n* = 8 for D46N and E133Q, *n* = 3 for R47Q). (C, D), pH dependency of SSME currents initiated by galactonate concentration jump (10 mM) at symmetrical pH conditions. Experiments were performed on *n* = 4 (C), *n* = 5 (D). The error bars represent standard deviations. (E) Concentration-dependent changes in $k_{obs}$, obtained by monoexponential fit of the current decay, for WT and mutant DgoT. Solid lines are fits to a three-states induced-fit model given above the graph. Experiments were performed on *n* = 7 (WT), *n* = 4 (E133Q), *n* = 4 (R126Q) and *n* = 3 (D46N) independent sensors. The error bars represent standard deviations. (F, G) pH dependency of SSME currents initiated by galactonate concentration jump (10 mM) at symmetrical pH conditions. Experiments were performed on *n* = 5 (F) and *n* = 2 (G) independent sensors. The error bars represent standard deviations. (C, D, F, G) Insets show pH dependences of normalized peak currents for each mutant. (H) Substrate concentration dependence of the peak current values for WT and R47Q DgoT. R47Q concentration dependencies were fitted to a two-state binding model given above the graph. Values of peak current values for WT DgoT (fitted to three-states model from (E)) are given for comparison. Experiments were performed on *n* = 5 (WT) and *n* = 4 (R47Q) independent sensors. The error bars represent standard deviations. Source data are available online for this figure.

probability (Fig. 7D). Given that substrate binding and release are fast processes compared with the conformational changes, our data show that reorientation of the empty transporter upon substrate release is the rate-limiting step in the transport cycle.

## Neutralization of putative proton acceptors abolishes galactonate transport

To further investigate the function of the charged transmembrane residues, we next evaluated the effects of the corresponding

neutralization mutations (D46N, E133Q, R47Q, and R126Q) on DgoT transport activity. We first monitored changes in the extracellular pH of a bacterial suspension using a micro pH electrode-based transport assay (Bosshart et al, 2019). The addition of galactonate to *E. coli* expressing WT DgoT resulted in a time-dependent increase in pH, which reached the maximum value in less than a minute. The addition of the same concentration of the galactonate epimer gluconate to DgoT-expressing *E. coli* nor of galactonate to vector-transformed (without DgoT) bacteria induced an increase in pH (Fig. 8A). All mutations tested abolished the galactonate-induced pH changes (Fig. 8B), indicating loss of transport function.

In SSME experiments, the application of galactonate elicited a positive fast pre-steady state, but not transport currents, for D46N and E133Q DgoT, indicating that these mutant transporters can bind galactonate but cannot complete the transport cycle. Galactonate-induced pre-steady state currents recorded with the E133Q mutant were pH independent (Fig. 8C), suggesting that protonation of this residue is responsible for the inhibition of WT DgoT at alkaline pH. At every pH tested, E133Q currents closely resembled WT currents under acidic pH conditions (Appendix Fig. S12A), where transport is blocked by impaired proton release inside the vesicle. In contrast, D46N currents were pH dependent, with the largest peak currents observed under acidic pH conditions (Fig. 8D). Currents obtained with this mutant were biphasic (Appendix Fig. S12B) and with slower decay than for E133Q. To determine whether the recorded currents represent the pre-steady state reaction or residual transport activity, we compared the time courses of current decay using liposomes reconstituted with different LPRs. Unlike the currents recorded with WT protein, the decay times for D46N DgoT did not systematically depend on the LPR (Appendix Fig. S12C). Therefore, the observed reaction can be attributed to a slow pre-steady state process.

For D46N and E133Q DgoT, the peak current amplitudes changed with increasing galactonate concentrations in a saturating fashion (Appendix Fig. S13), suggesting that these mutant transporters undergo conformational changes after rapid galactonate binding (Bazzone et al, 2021). Similar behavior was observed for WT DgoT under acidic conditions, where only the pre-steady state reaction (no transport activity) can be seen (Figs. 2A and EV2A). Figure 8E shows that the observed rates (obtained as reciprocal decay time constants) depend on the galactonate concentration. Hyperbolic curve fitting provided the $K_D$s for galactonate binding, as well as the kinetic parameters characterizing the conformational changes (Table 1). Compared with WT, conformational changes are similar for E133Q and slower for D46N, but with no difference in substrate affinity. We conclude that deprotonation of D46 and E133 is necessary to complete the transport cycle; however, partial reactions that represent conformational changes can occur in the presence of galactonate.

## R47 couples substrate binding and major conformational changes

For neutralizing mutations of both transmembrane arginine residues (R47Q and R126Q), galactonate application elicited fast negative currents in SSME experiments (Fig. 8F,G). For R126Q, current amplitudes were similar at pH 5–7, but lower at more alkaline pH. Peak currents changed with increasing galactonate

**Table 1. Overview of maximum peak current values and kinetic parameters for WT DgoT and transport-deficient mutants.**

| Variant | $I_{max}$ (nA) | $k_{obs}$ (s$^{-1}$) | $K_D$ (mM) | $k^-$ (s$^{-1}$) | $k^+$ (s$^{-1}$) |
|---|---|---|---|---|---|
| WT at pH 5.8 | 0.7 ± 0.2 | 79 ± 14 | 15.1 | 23.8 | 70.6 |
| D46N | 2.7 ± 0.3 | 65 ± 8 | 10.9 | 6.6 | 69.9 |
| E133Q | 1.0 ± 0.3 | 105 ± 15 | 15.3 | 30.8 | 92.2 |
| R47Q | −0.37 ± 0.05 | | | | |
| R126Q | −1.5 ± 0.4 | 74 ± 11 | 22 | 14.4 | 86.7 |

Values were averaged over $n = 3$ different sensors (errors reflect standard deviations). $I_{max}$ and $k_{obs}$ were measured upon addition of 50 mM galactonate. For R47Q analysis of current decays was limited by time resolution of the instrument, therefore only $I_{max}$ values are given.

concentrations with a hyperbolic concentration dependence (Fig. 8E), indicating conformational changes. In contrast, R47Q DgoT exhibited faster currents with amplitudes increasing linearly with increasing substrate concentration (Fig. 8H). Therefore, the observed pre-steady state reaction for R47Q differs from those observed for other transport-deficient mutants and likely represents electrogenic substrate binding rather than conformational changes (Bazzone et al, 2021).

The crystal structure of outward-facing DgoT (Leano et al, 2019), as well as our MD simulation data (Fig. 3), indicates that R47 interacts with the carboxyl group of bound galactonate. To better understand the role of this arginine residue, we carried out MD simulations with the outward-facing DgoT structure in which R47 was mutated to glutamine and D46 and E133 were protonated. In *apo* simulations, both WT and mutant DgoT assumed similar conformations with an open extracellular gate (Fig. 9A). However, galactonate failed to promote closure of the extracellular gate of R47Q DgoT (Fig. 9B). This difference is a consequence of changes in protein–substrate interactions. Galactonate is coordinated by multiple polar residues (Batarni et al, 2023) in helices from both the N- and C-domains, with the substrate located closer to TM7 in simulations with the mutant than with the WT transporter (Fig. 9C,D). Comparison of the substrate–protein interactions revealed more stable contacts for galactonate with C-domain residues such as Q264, T372, and N393 in WT; residue 47 from the N-domain interacts with the substrate only in simulations with the WT protein, but not the R47Q mutant (Fig. 9E).

Our combined experimental and computational results describe substrate binding as a multiphasic process initiated by galactonate recognition and accommodation in the binding site. Subsequent direct interaction between the carboxyl group of galactonate and R47 induces closure of the extracellular gate. Both reactions are electrogenic; R47Q abolishes the second step without affecting the initial substrate binding. The fast negative component corresponding to electrogenic galactonate binding also occurs with other mutants, along with an additional slower component of positive (E133Q) or negative (R126Q) amplitude (Fig. 8C,F).

## Discussion

We combined experimental and computational approaches to describe the transport cycle of the bacterial SLC17 homolog DgoT. DgoT transport is based on an alternating access mechanism, with

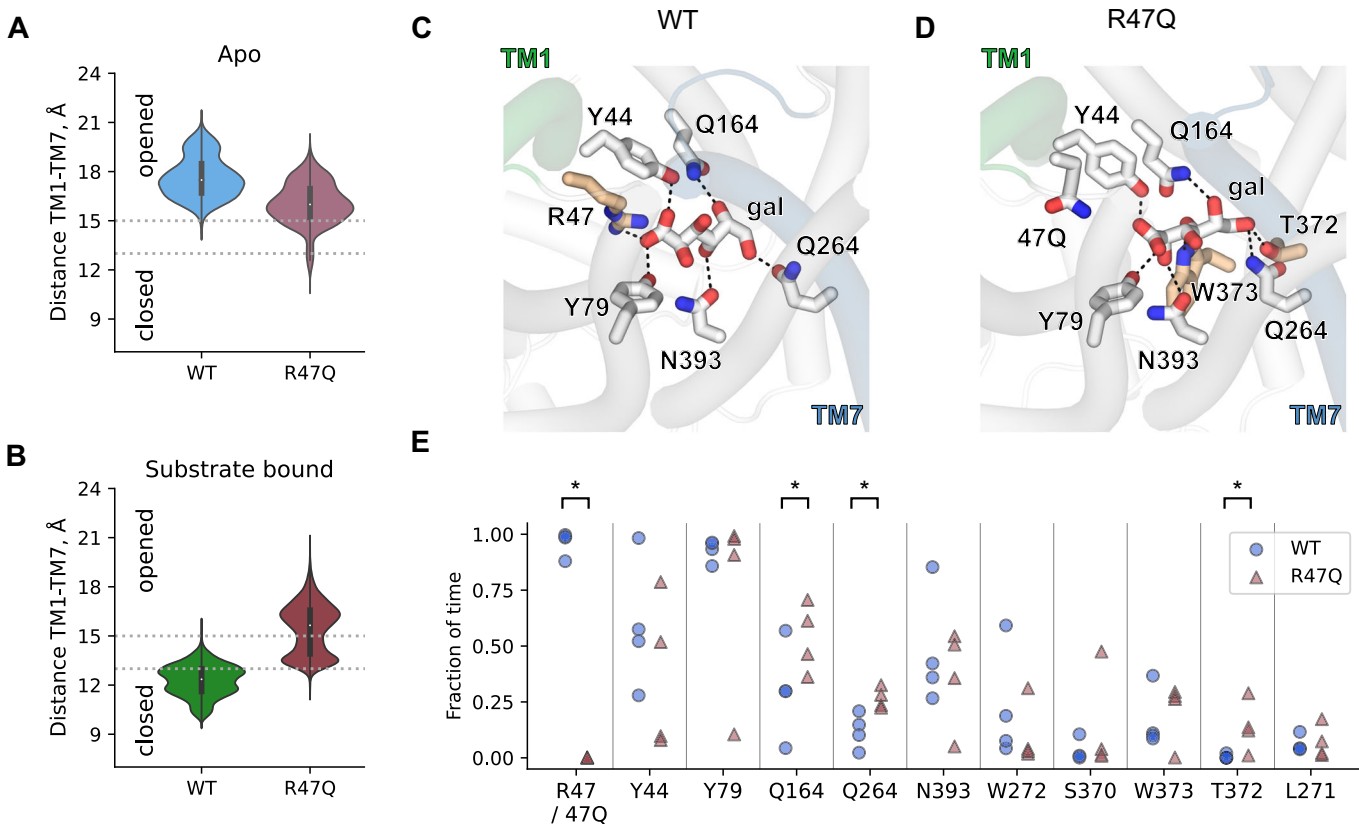

**Figure 9. Galactonate does not induce extracellular gate closure in R47Q DgoT.**

(A, B) Probability densities for extracellular gate opening in *apo* (A) and galactonate-bound (B) simulations with WT and R47Q DgoT in outward-facing conformation with D46 and E133 protonated (*n* = 4 for each condition). The central white dots represent the medians, thick black lines represent ranges between the 25th percentile (Q1) and the 75th percentile (Q3), the top and bottom points of the violin extend to the minimum and maximum values of the kernel density estimate (KDE). (C, D) Position of galactonate in the binding site of WT (C) and R47Q DgoT (D). (E) Fraction of frames, in which distance between the side chain of specified residue and closest oxygen atom of galactonate was equal or less than 2 Å. Each data point represents one of the four independent trajectories for WT (blue points) and R47Q mutant (red points) DgoT. Significance was evaluated with the Mann–Whitney test, one-sided: *$P < 0.05$ ($P = 0.013$ for R47, $P = 0.049$ for Q164, $P = 0.014$ for Q264, $P = 0.027$ for T372). Source data are available online for this figure.

the transporter cycling between inward- and outward-facing states through occluded conformations in which the substrate-binding site is inaccessible from both membrane sides (Fig. 10). In the outward-facing conformation, DgoT dynamically switches between states with a closed or open extracellular gate (states 1 and 2). Protonation of both D46 and E133 stabilizes the open-gate conformation (state 2) and permits galactonate binding from the periplasmic side (state 3), followed by closure of the extracellular gate. Formation of the outward-occluded conformation is the initial step of a major conformational change (state 4) that brings the protein into an inward-facing conformation (states 5, 6, and 7). In this conformation, deprotonation of D46 (state 5) opens the intracellular gate that permits galactonate release (state 6). Deprotonation of D46 may occur via multiple pathways, either stepwise or concerted. Our multiscale simulations demonstrate that one of such pathways involves initial proton release from E133, either to the intracellular solvent or to galactonate, followed by H$^+$ transfer from D46 to E133 (state 6). However, further characterization with two-dimensional free energy sampling and improved collective variables for proton transfer (Li and Voth, 2021b), as well as complementary

computational approaches, such as adaptive QM/MM schemes (Bulo et al, 2009) or multistate reactive MD (Kaiser et al, 2024), will be necessary to fully understand this and other putative proton release pathways. After D46 deprotonation, galactonate can unbind in a protonated or non-protonated form, though protonation might promote galactonate release by weakening the electrostatic interaction with R47 and stabilizing intracellular gate opening. Protonated galactonate could release the proton, either once it reaches the internal bulk (as its pK$_a$ in solution is 3.39) or while it is leaving DgoT. Although we cannot predict if and where galactonate deprotonates along the dissociation pathway, it is unlikely that galactonate release and/or deprotonation are the rate-limiting step of the proton/substrate release mechanism, as dissociation was observed in unbiased classical MD simulations for both protonated and deprotonated galactonate (Appendix Table S1). After H$^+$ transfer from D46 to E133 (state 6), subsequent proton release from E133 results in intracellular gate closure (state 7). Finally, reorientation of the empty transporter to the outward-facing conformation completes the cycle (state 8); this last step appears to be the rate limiting of the transport cycle.

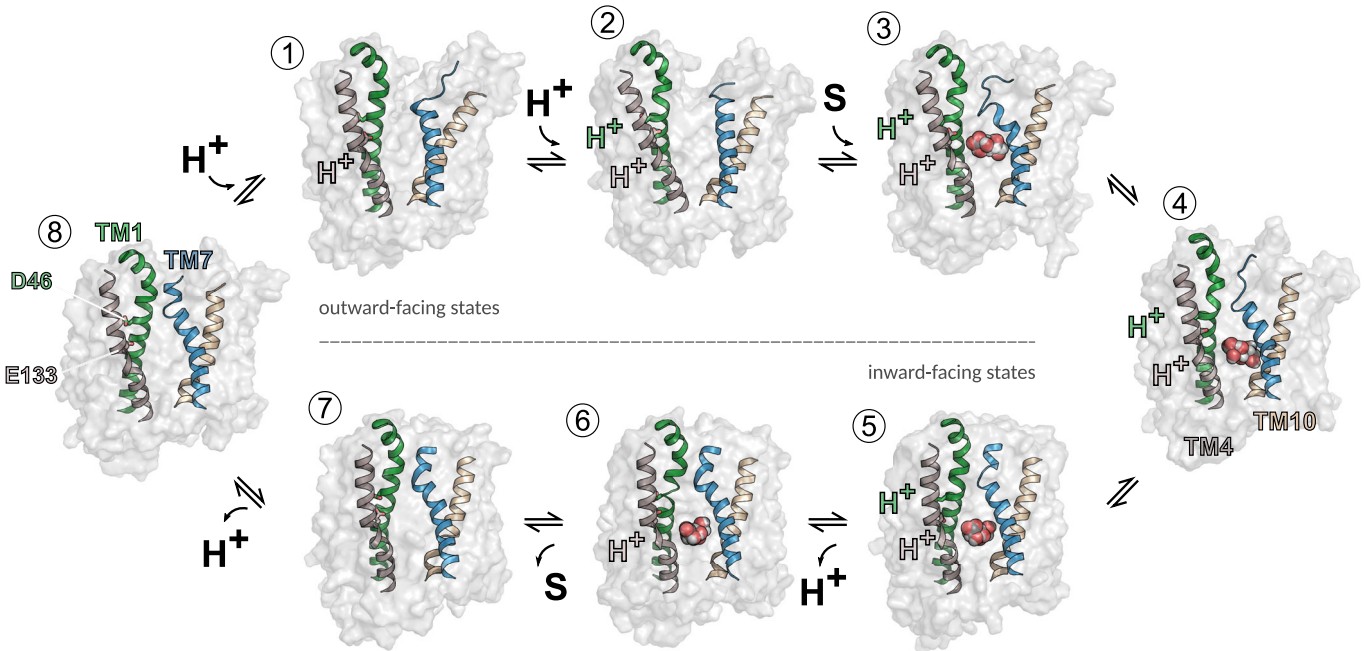

**Figure 10. Schematic transport cycle of DgoT.**

In outward-facing conformation (1), extracellular gate opening is stabilized by protonation of D46 and E133 (2). Galactonate binding (3) induces major conformational changes (4). In the inward-facing state (5), proton release from D46 leads to galactonate release to the intracellular solution (6), followed by the release of the second proton (7) and reorientation of the empty carrier to outward-facing conformation (8).

Our classical MD simulations revealed how the protonation state of the two acidic transmembrane residues is coupled to the conformational changes necessary for substrate binding and release. In the *apo* state, deprotonated D46 and deprotonated E133 both serve as "latches" by interacting with R126 (TM4) and R47 (TM1), respectively; this strategy is commonly used by MFS transporters (Drew et al, 2021). Protonation of the two acidic residues unclasps these latches, albeit with distinct effects on extracellular gate mobility (Figs. 3D and 5D). For D46, H⁺ association releases R126 and shifts the equilibrium toward a state with an open extracellular gate, whereas protonation of E133 alone stabilizes the gate in an only slightly opened state. Double protonation is necessary for full opening of the extracellular gate, which makes substrate binding most effective (Figs. 3D and 4D). Galactonate binding promotes closure of the extracellular gate for all four protonation state combinations (Fig. 3E). However, stable gate closure requires protonation of E133 (Fig. 3E), which disrupts the E133–R47 salt bridge and gives TM1 the flexibility to come into closer contact with TM7 upon galactonate binding (Fig. 3C).

Galactonate application in experiments with purified DgoT does not result in H⁺ binding or release (Fig. 2F). This result is in agreement with galactonate binding to the transporter after protonation of D46 and E133. In these experiments, there is no driving force for subsequent proton release, since there is no lipid bilayer and thus proton or substrate concentration gradients are lacking Together with our Markov state modeling indicating that the *apo* transporter predominantly assumes the outward-facing conformation (Fig. 7), these experimental results show that protonation of both D46 and E133 precedes galactonate binding from the periplasmic side. At present, we cannot exclude a certain

fraction of *apo* DgoT is present in the inward-facing conformation. If there existed a significant fraction in this state, one would have to conclude that DgoT assumes the same binding order in forward and in reverse transport.

Arginine residues at positions 47 and 126 are conserved within SLC17 transporters (Fig. EV1; Appendix Fig. S1). R47 forms an electrostatic interaction with galactonate in the binding site that is suggested to be responsible for its recognition (Leano et al, 2019). However, R47Q DgoT exhibits galactonate-specific pre-steady state currents (Fig. 8G) that represent a binding reaction rather than rate-limiting conformational changes. Therefore, our results assign a role beyond substrate recognition to R47: this arginine connects the substrate-binding site to the protonation site and triggers closure of the extracellular gate, much like H322 in lactose permease LacY (Kumar et al, 2015). Pre-steady-state currents of R126Q DgoT indicate electrogenic conformational changes rather than a binding reaction. R126 might be involved in tuning the pKₐ of D46 at different stages of the transport cycle or be necessary for conformational changes in *apo* DgoT.

The proposed transport mechanism (Fig. 10) fully agrees with all published experimental data. It accounts for the stoichiometrically coupled symport of two H⁺ and one galactonate, as determined experimentally (Fig. 2E). Solubilized DgoT does not bind protons after the addition of galactonate (Fig. 2F), as would be expected if protonation of the transporter precedes substrate binding. Neutralization of residue D46 or E133 prevents coupled galactonate transport, as expected for the removal of obligatory protonation sites. Our transport scheme can also account for recently reported experiments demonstrating that E133Q, but not D46N, DgoT is capable of galactonate exchange (Batarni et al, 2023). The E133Q mutation has no effect on galactonate release and

substrate exchange: it exclusively abolishes forward transport by preventing *apo* translocation back to the outward-facing conformation. In contrast, D46N prevents forward transport as well as galactonate exchange, since deprotonation of D46 is required for galactonate release (Fig. 5). The surprising result that the additional E133Q mutation restores galactonate exchange in D46N/E133Q DgoT might be explained if galactonate is predominantly released in a protonated form by WT DgoT. In D46N DgoT, galactonate may still be protonated via $H^+$ transfer from E133 although the closed intracellular gate would prevent its release (Fig. 5D). In D46N/E133Q DgoT, galactonate cannot be protonated and the slightly higher open probabilities for the intracellular gate with non-protonated galactonate make the double mutant competent for exchange.

*E. coli* can use galactonate as the sole carbon source (Deacon and Cooper, 1977), demonstrating that DgoT transports as effectively as alternative glucose transporters, such as lactose permease LacY (Kaback and Guan, 2019) or xylose symporter XylE (Madej et al, 2014). LacY and XylE transport uncharged sugar molecules with a stoichiometry of 1 proton:1 substrate, whereas DgoT co-transports two protons with the negatively charged galactonate, together resulting in a positive net transported charge. Most likely, the adjusted transport stoichiometry permits DgoT to utilize $\Delta pH$ and $\Delta\Psi$ and provides high driving forces, thus optimizing bacteria for nutrient uptake when resources are limited.

Our finding that galactonate can be released as protonated substrate and thus act as carrier of one of the two protons needed for transport is not unexpected, as substrate titration was already reported for another member of the MFS family, the phosphate transporter PiPT (Liu et al, 2021; Liu et al, 2024). Therefore, despite the different $pK_a$ values of galactonate and phosphate (3.39 and 7.21) and different number and distribution of titratable residues in DgoT and PiPT, both transporters exploit proton transfer to the titratable substrate to drive the transport cycle. The use of kinetic network modeling (Parker et al, 2017; Liu et al, 2021; Swanson, 2022; Li et al, 2022; Liu et al, 2024) would be needed to investigate whether protonated galactonate can release the proton while it is leaving the DgoT permeation pathway or when it reaches the intracellular solution.

The main habitat of *E. coli* is the mammalian intestinum, where it is exposed to neutral and alkaline pH in the small intestine or to acidic pH between pH 5.5 to 6.5 in the colon (Yamamura et al, 2023). For acidic and neutral external pH, an inwardly directed pH gradient (Slonczewski et al, 1981) provides the driving force for coupled 2 $H^+$:1 galactonate transport of DgoT, resulting in fast and efficient galactonate accumulation.

Transport substrates and stoichiometries vary substantially among MFS transporters. The lactose permease LacY (Kaback and Guan, 2019) and fucose transporter FucP (Dang et al, 2010) share a stoichiometry of 1 $H^+$:1 substrate. In LacY, protonation of a single site, E325 (equivalent to E133 in DgoT), permits lactose binding and transition to the occluded conformation, as observed upon double titration of D46 and E133 in DgoT. FucP has two protonation sites, D46 and E135, that are homologous to D46 and E133 in DgoT. However, in FucP only D46 is accessible from the extracellular solution, with E135 serving only as part of the proton transfer pathway. SLC17A5/sialin differs from DgoT in having broad substrate specificity: it recognizes various sialic acids via electrostatic interaction with two conserved arginines, R168 and R57 (Hu et al, 2023). Two glutamates, E171 and E175, serve as protonation sites. Protonation of E171 releases R168 and permits its interaction with the substrate. Subsequent $H^+$ transfer to E175 is followed by substrate transfer to a more cytoplasmic binding site close to R57 and translocation to the inward-facing conformation. Such coupling of substrate translocation to proton transfer results in the electroneutral co-transport of one $H^+$ and one sialic acid molecule.

The comparison of DgoT with FucP and SLC17A5/sialin thus reveals how small variations in the arrangement of protonation sites can adjust transport stoichiometries of $H^+$-coupled transporters with conserved architecture and transport mechanisms. Besides DgoT, SLC17 family encompasses organic anion uniporters (Ishikawa et al, 2008; Iharada et al, 2010), $H^+$-glutamate exchangers (Kolen et al, 2023), and the aforementioned $H^+$-sialic acid symporter (Hu et al, 2023). Certain SLC17 transporters can also function as a $Na^+$-$PO_4^{3-}$ symporter (Iharada et al, 2010; Preobraschenski et al, 2018). Therefore, our work may serve as a framework to understand the mechanisms underlying the diversity of SLC17 transport mechanisms.

# Methods

**Reagents and tools table**

| Reagent/resource | Reference or source | Identifier or catalog number |
|---|---|---|
| **Experimental models** | | |
| N/A | | |
| **Recombinant DNA** | | |
| DgoT WT | *E. coli* | N/A |
| **Antibodies** | | |
| N/A | | |
| **Oligonucleotides and other sequence-based reagents** | | |
| PCR primers for WT DgoT (5′–3′ sequences): | | |
| S Dgot | Eurofins Genomics | AAAAAACCATGGAT GGTGAGCGGCTTCGCTATG CCCAAAATC |
| AS DgoT | Eurofins Genomics | TTTTTTAAGCTTTTAATGGTG ATGATGGTGATGATGGTGA TGATGGCTGCCGCGCGGC ACCAGGCCAACGCGCTTC ACATCGCCCACCAGCAG |
| **Chemicals, enzymes and other reagents** | | |
| Calcium D-galactonate | Biosynth | FC57272 |
| Sodium D-galactonate | Leano et al, 2019 | N/A |
| **Software** | | |
| SURFE²R N1 Control | Nanion Technologies | N/A |
| GROMACS versions 2018, 2020 and 2021 | https://manual.gromacs.org/ | N/A |
| CPMD version 4.3 | Hutter et al, 2000 | N/A |
| MiMiC version 0.2.0 | Olsen et al, 2019; Bolnykh et al, 2019 | N/A |

| Reagent/ resource | Reference or source | Identifier or catalog number |
|---|---|---|
| PLUMED version 2.8.1 | Tribello et al, 2014 | N/A |
| PyMOL version 2.5.0 | https://pymol.org/2/ | N/A |
| g_elpot | https://jugit.fz-juelich.de/ computational-neurophysiology/g_elpot | N/A |
| Python 3.7, 3.9, 3.11 | https://www.python.org/ | N/A |
| Jalview | https://www.jalview.org/ | N/A |
| ClustalW | http://www.clustal.org/ clustal2/ | N/A |
| Other | | |
| SURFE²R N1 Control | Nanion Technologies | N/A |

## Expression and protein purification

The full-length DgoT gene (GenBank accession number AKK15832.2) was subcloned into a pQE60 vector through the NcoI and HindIII restriction sites in fusion with a C-terminal thrombin cleavage site and decahistidine tag. Mutant constructs were generated using PCR-based mutagenesis and verified by DNA sequencing. Protein expression and purification were performed using a published procedure, with modifications (Leano et al, 2019). *E. coli* C41 cells transformed with pQE60 DgoT WT were grown at 37 °C in TB medium supplemented with 2 mM MgSO$_4$. When an OD$_{600}$ of 0.6–0.8 was reached, gene expression was induced with 1 mM IPTG and cells were grown for a further 4 h (typical yield of 15 g per 1 L culture). After sedimentation, cells were flash-frozen in liquid nitrogen and stored at −80 °C for later use, Next, cells (15 g) were resuspended in 20 mM Tris (pH 7.4) and 300 mM NaCl (50 mL volume) containing complete protease inhibitor cocktail (Roche) and lysed by sonication. Debris was removed by centrifugation at 12,000 × $g$ for 15 min, and membranes were collected at 200,000 × $g$ for 1 h, flash-frozen in liquid nitrogen, and stored at −80 °C until use.

A frozen membrane pellet from 15 g cells was resuspended in 20 mL membrane buffer (20 mM Tris (pH 7.4), 150 mM NaCl) containing cOmplete protease inhibitor cocktail, using a glass Dounce homogenizer, and then *n*-dodecyl-D-maltoside (DDM) was added to 1.4%. Membranes were solubilized for 2 h at 4 °C and the insoluble fraction was removed by ultracentrifugation at 75,000 × $g$ for 30 min at 4 °C. The supernatant was diluted 1:2 with solubilization buffer, imidazole was added to 15 mM, and pH was adjusted to 7.8–8.0. CoNTA resin (3 mL) was washed with 10 column volumes (CV) of membrane buffer, added to the super-natant, and incubated at 4 °C for 1.5 h under gentle agitation. The resin was then washed with 10 CV of wash1 buffer (20 mM Tris, 150 mM NaCl, 15 mM imidazole, 0.1% DDM, pH 8.0) and 10 CV of wash2 buffer (20 mM Tris, 150 mM NaCl, 0.1% lauryl maltose neopentyl glycol (LMNG), pH 7.4). Protein was eluted with 4 CV of elution buffer (20 mM Tris, 150 mM NaCl, 0.1% LMNG, 150 mM imidazole, pH 7.4) into 0.5 CV fractions, and 5 mM EDTA was added immediately after collection. Protein concentration in the eluted fractions was estimated by measuring the absorbance at

280 nm (NanoDrop), and fractions with the highest protein concentrations were combined. Imidazole was removed using a PD-10 desalting column, with protein eluted with buffer (20 mM Tris, 150 mM NaCl, 1 mM EDTA, 0.05% LMNG, pH 7.4) and stored overnight at 4 °C.

The purified protein was concentrated to 5–8 mg/ml using a 50 kDa molecular weight cutoff (MWCO) centrifuge concentrator (Millipore) and loaded in 0.5 mL portions onto a Superdex 200 Increase 10/300 GL column (GE Healthcare Life Sciences) preequilibrated with size-exclusion chromatography (SEC) buffer (20 mM HEPES, 150 mM NaCl, 0.05% LMNG, pH 7.4). The peak fractions were combined, flash-frozen in liquid nitrogen, and stored at −80 °C until use.

## Reconstitution of proteoliposomes

*E. coli* polar lipids (Avanti Polar Lipids, *E. coli* polar lipid extract, 25 mg/ml solution in chloroform) were dried under nitrogen and then under vacuum overnight. The dried lipid film was resuspended to a lipid concentration of 10 mg/ml by stirring in reconstitution buffer (1 mM HEPES pH 7.4, 150 mM NaCl, 2 mM MgSO$_4$) for 1 h at room temperature (RT). After the lipids were completely dissolved, the suspension was frozen in liquid nitrogen, stirred for another 1 h at RT, and sonicated using a UP50H ultrasonic processor equipped with a microtip until clear (2 or 3 cycles of 30 s). The formed liposomes were destabilized by adding 0.6% Triton X-100 and incubated for 45 min at RT under gentle agitation. Purified DgoT was added to the destabilized liposomes at a LPR of 5, 10, or 20 and then incubated for 1 h at 4 °C. To remove the detergent, 150 mg SM2 Bio-beads was added per 1 mL liposome suspension. After incubation for 1 h at 4 °C, another 150 mg Bio-beads was added per 1 mL suspension, followed by another 1 h incubation at 4 °C, and then the beads were removed using a disposable column. A third volume of Bio-beads was then added (400 mg per 1 mL suspension), incubated overnight at 4 °C, and removed using a disposable column. Empty liposomes were prepared in parallel using the same procedure but with no added protein.

## pH electrode-based transport measurements

DgoT was expressed in *E. coli* C41 cells as described for protein purification. After protein expression, cells were pelleted, washed twice in assay solution (250 mM KCl, 1 mM MgSO$_4$, 2 mM CaCl$_2$), and resuspended in assay solution to an OD$_{600}$ 15. All centrifugations were performed at 2200 × $g$ for 7–8 min, and cells were resuspended by vortexing. Substrates to be assayed were dissolved in assay solution at a concentration of 160 mM and the pH was adjusted to 6.5–6.7 with KOH such that the substrate pH was lower than the pH of the bacterial suspension for each experiment. A 800 μL volume of bacterial suspension was transferred to a 2 ml reaction tube (Eppendorf) and the pH of the suspension (i.e., of the extracellular medium) was measured using a micro pH electrode with an integrated temperature sensor (Xylem, SI Analytics) under constant stirring and adjusted to pH 6.7 using KOH and HCl. After recording the baseline for 60 s, 50 μL of the corresponding compound was added to the bacterial suspension (final concentration: 10 mM). All experiments were performed at 20–21 °C. For experiments with DgoT in detergent, purified DgoT (stored at −80 °C) was thawed and SEC buffer was

replaced using a PD-10 desalting column with assay solution containing 150 mM NaCl and 0.05% LMNG (pH ~7.4). Purified protein was used at a concentration of 4 µM in all experiments. The experiment was repeated six times, with comparable results.

## SSM-based electrophysiology

Gold electrode sensors (1 or 3 mm) were prepared as previously described (Bazzone et al, 2017). Briefly, sensors were incubated for at least 30 min in an octadecane thiol solution, and then rinsed thoroughly with isopropanol and water. The solid-supported membrane (SSM) was prepared by pipetting 1.5 µL diphytanoyl phosphatidylcholine dissolved in *n*-decane onto the electrode surface, followed by 100 µL aqueous buffer. Immediately prior to measurements, liposome samples were thawed, diluted to a final lipid concentration of 1 mg/mL and briefly sonicated. Each liposome sample (10 µL) was pipetted onto a SSM sensor and adsorbed by centrifugation at $2200 \times g$ for 30 min at RT. All experiments were repeated for at least three sensors, with each condition measured at least twice. All solutions were buffered in 100 mM potassium phosphate (KPi) for each pH used.

For measurements with a single solution exchange protocol, three phases of 1 s duration were applied: flow of nonactivating (NA) solution, activating (A) solution, and NA solution. Only the A solution contained galactonate. In experiments with variable pH, pH of NA and A solutions was kept constant within a single experiment. Between experiments using different pH values, the sensor was incubated at the new pH for 5 min to equilibrate the intraliposomal pH. In experiments with WT, D46N, E133Q, and R126Q DgoT, the NA solution contained gluconate to compensate for galactonate in the A solution. For R47Q DgoT, glutamate was used in NA solution instead, allowing a direct comparison of the responses to galactonate and gluconate applications.

To determine the apparent $pK_a$ values, normalized peak currents (measured using 3 mm sensors) were fitted with one of the following equations:

$$\text{acidic deactivation}(pH \leq 8) : I_{norm}(pH) = \frac{I_{max}}{1 + 10^{pK_a - pH}};$$

$$\text{alkaline deactivation}(pH \geq 7.5) : I_{norm}(pH) = \frac{I_{max}}{1 + 10^{pH - pK_b}}.$$

Currents were normalized to the $I_{max}$ value obtained by the fit of peak currents measured on the same sensor.

For the analysis of pre-steady state currents, 1 mm sensors were used. Rate constants for the observed charge displacements ($k_{obs} = 1/\tau$) were derived from the transient currents by fitting the decay with a monoexponential function $I = A * \exp(-t/\tau)$. This two-step reversible reaction can be described as follows:

$$P + S \underset{K_D}{\longleftrightarrow} PS \underset{k^-}{\overset{k^+}{\rightleftarrows}} P^*S,$$

where $P$ is the protein, $S$ is the substrate, and $P^*$ is the protein after conformational changes. The first step is the binding reaction described by the dissociation constant $K_D$, and the second step is the substrate-induced conformational change characterized by the forward and reverse rate constants. Assuming that substrate binding is rapid, the observed rate constant has hyperbolic dependence on

the substrate concentration (Smirnova et al, 2006):

$$k_{obs} = k^- + k^+ \frac{[gal]}{[gal] + K_D}.$$

Measurements under asymmetrical pH conditions were done using a double solution exchange configuration. A resting (R) solution phase of 1 s and incubation period of 5–20 min were added to the beginning of each measurement to allow the intraliposomal pH to adjust to the pH of the R solution. Afterwards, a normal single solution exchange protocol was used to establish the pH gradient (during NA phase) and substrate gradient (A phase).

To determine the transport stoichiometry, we used a reversal assay as previously described (Thomas et al, 2021), with a single solution exchange protocol comprising three phases (NA, A, and NA) extended to 2, 3, and 3 s, respectively. Between measurements, the sensor was rinsed 3–5 times with NA solution; the current responses were recorded and used as a baseline. The same protocol was used for samples with empty liposomes to account for solution exchange artifacts. The entire A phase was integrated to obtain the transported charge values. After subtraction of the negative control (integrated current recorded with empty liposomes), the transported charge values were used to determine the transport stoichiometry. The internal (NA) solution used for this experiment had a pH of 7.3 and contained 0.5 mM galactonate and 7.5 mM gluconate, and applied external (A) solutions had a pH of 7.6 and contained X mM galactonate and 8–X mM gluconate. This setup resulted in $d\mu_{gal}/d\mu_{H+}$ values of 0–4.

## Classical MD simulations

DgoT protein coordinates obtained from the Protein Data Bank (inward-facing, PDB ID: 6E9N; outward-facing galactonate-bound, PDB ID: 6E9O (Leano et al, 2019)) were used as the starting coordinates for MD simulations. The D46 and E133 protonation state and substrate occupation were modified as described in the text. Standard protonation states at neutral pH were assigned to all other residues (deprotonated aspartate and glutamate residues, and epsilon protonated, neutral histidine residues), except for H56, which forms a salt bridge with E180 and, therefore, was set as doubly protonated. Missing residues (235–242 in the inward-facing structure, 231–243 and 277–290 in the outward-facing structure) were modeled using the SWISS MODEL server (Waterhouse et al, 2018) with a final residue range of 27–442 used for both protein structures. The N- and C-termini were capped with neutral acetyl and methylamide groups, respectively. For modeling DgoT single point mutations, we used PyMOL (Schrödinger, LLC, 2015). Glutamine residue in position 133 in the outward-facing crystal structure of the E133Q mutant DgoT was replaced with glutamate to model the WT protein prior to MD equilibration.

The initial protein orientation within the membrane was set to the corresponding DgoT structure available in the Orientations of Proteins in Membranes database (Lomize et al, 2012). The protein was then embedded into a phosphatidylcholine (POPC) bilayer using *g_membed* (Wolf et al, 2010) in GROMACS and solvated in a box with dimensions $\sim 120 \times 120 \times 100$ Å, which was chosen to ensure a minimum distance between periodic copies of at least 30 Å. The simulation temperature was 310.15 K. The protein/

membrane system was surrounded by a ~100 mM solution of $Na^+$ and $Cl^-$ ions. Ions were described using default CHARMM parameters, and the CHARMM TIP3P model was used for water molecules. MD simulations were performed using the GROMACS software package (versions 2018, 2020, and 2022) (Abraham et al, 2015) with the CHARMM36m force field (Klauda et al, 2010; Huang et al, 2017). Galactonate (deprotonated and protonated) parameters were obtained using the SwissParam server (Zoete et al, 2011) and added to the forcefield. An integration time step of 2 fs was used. Van der Waals interactions were calculated with the Lennard–Jones potential and a cutoff radius of 1.2 nm, with forces smoothly switched to zero in the range of 1.0–1.2 nm and no dispersion correction applied. Electrostatic interactions were calculated by the particle mesh Ewald method (Essmann et al, 1995), with a real-space cutoff distance of 1.2 nm. All simulations were done in an isothermal–isobaric ensemble, with the temperature set to 310 K using a v-rescale thermostat (Bussi et al, 2007) and a time constant of 0.5 ps. The thermostat was applied separately to the protein, lipid bilayer, and aqueous solution containing ions. The same groups were used for the removal of the center-of-mass linear motion.

The protein was equilibrated in three steps using the velocity-rescale thermostat and Berendsen (Berendsen et al, 1984) pressure coupling. The first step lasted 50 ns and was run with positional restraints on protein atoms with a harmonic potential with a force constant of 1000 kJ mol$^{-1}$ nm$^{-2}$ to allow for equilibration of water and ions. In the second step, only the backbone atoms of the protein were restrained to enable side chains to equilibrate for another 20 ns. Lastly, the system was equilibrated for 1 ns without positional restraints to obtain the velocities used in the following production runs. Production MD simulations used Parrinello–Rahman (Parrinello and Rahman, 1981) pressure coupling in a semi-isotropic manner with a time constant of 0.5 ps and a target pressure of 1 bar.

Besides unbiased classical MD, we ran additional simulations biasing the side chain of protonated D46 towards the *close* conformation for inward-facing, substrate-bound DgoT. Specifically, the D46 dihedral angle $\chi_1$ (defined by atoms NH1, CT1, CT2 and CD using the CHARMM naming convention) was restrained to a target value ($\chi_1^{target}$) of 72° with a harmonic potential with a force constant of 20 kJ mol$^{-1}$ nm$^{-2}$ for $|\chi_1 - \chi_1^{target}| > 18°$.

Information about simulations' lengths is provided in Appendix Table S3. To analyze convergence of the production runs, time-dependent overlap of unbiased probability distributions for the complete dataset (Appendix Fig. S3A,B) with those obtained for non-overlapping 10-ns blocks was determined (Appendix Fig. S3C).

Estimation of p$K_a$ values was performed using the PROPKA3 software package (Olsson et al, 2011).

## Classical WT-MTD simulations

The PLUMED v2.8.1 plugin (Tribello et al, 2014) interfaced with GROMACS v2020.4 (Abraham et al, 2015) was used for simulations, along with MD settings similar to those described for the unbiased MD simulations described in the previous section. The collective variable used to describe the conformational change of D46 is the N-Cα-Cβ-Cγ dihedral angle ($\chi_1$). WT-MTD parameters were used as previously reported (Chiariello et al, 2020; Chiariello

et al, 2021): Gaussian height, 1.2 kJ/mol; Gaussian sigma, 0.35 rad; bias factor, 15; and Gaussian deposition frequency, 500 MD steps. Employing a previously described algorithm (Tiwary and Parrinello 2015), we applied reweighting to the dihedral angle serving as the CV to reconstruct the histogram of the sampled configurations. WT-MTD simulations were run for 100 ns, which was sufficient to achieve a converged free energy surface (as shown in Appendix Fig. S14).

## QM/MM simulations

QM/MM simulations were carried out using GROMACS v2020.4 (Abraham et al, 2015) and CPMD v4.3 (Hutter et al, 2000), coupled with MiMiC v0.2.0 (Olsen et al, 2019; Bolnykh et al, 2019). MiMiC-QM/MM input files were generated using MiMiCPy v0.2.1 (Raghavan et al, 2023). Different QM partitions were used to investigate the different processes. Region QM0 was used to model protonation of D46 by E133 in outward-facing, *apo* DgoT and thus comprised the side chains of E133 and D46, as well as the nearby R126, which is hydrogen bonded to the latter acidic residue. The initial QM/MM configuration was extracted from one of the classical MD replica simulations of *apo* DgoT with protonated E133 and deprotonated D46. To simulate D46 deprotonation in inward, substrate-bound DgoT, we used different QM partitions, depending on the process under study. Region QM1 was used to simulate proton transfer from protonated E133 to the immediate neighbor water molecule and encompassed the side chain of E133, the galactonate molecule, and the three water molecules bridging these. The initial QM/MM configuration was obtained from WT-MTD simulations of the system in which both D46 and E133 were protonated. A snapshot in which E133 and galactonate were connected through water molecules was identified by visual inspection. Region QM2 included the hydronium ion formed by proton release from E133, the substrate galactonate, and the first layer of surrounding water molecules, enabling the simulation of proton migration to either galactonate or solvent. The initial QM/MM configuration was taken from the window corresponding to the products (CV1 = 0.35 Å) of the first TI proton transfer simulation (see next section). Lastly, QM3 included the side chains of E133 and D46 to enable the study of proton transfer from protonated D46 to deprotonated E133. Here, the initial QM/MM configuration was extracted through visual inspection from a classical MD simulation of the system with protonated D46 and now-deprotonated E133, in order to identify a snapshot with D46 in the *close* conformation and H-bonded to E133. The total number of QM atoms for each partition was 34 (QM0), 44 (QM1), 82 (QM2), and 18 (QM3).

The QM problem was solved using DFT (Hohenberg and Kohn, 1964; Kohn and Sham, 1965) with the BLYP functional (Becke, 1988; Lee et al, 1988). Core electrons were described using norm-conserving pseudopotentials of the Martins−Troullier type (Troullier and Martins, 1991), and the valence electrons were treated explicitly, using a plane wave basis set with cutoff 70 Ry. Monovalent carbon pseudopotentials were used to saturate the dangling bonds at the boundaries between the QM and MM regions (von Lilienfeld et al, 2005), with link atoms placed at the Cα atom of D46 and E133 and Cβ atom of R126. Isolated system conditions were achieved using the method of Martyna and Tuckerman (Martyna and Tuckerman, 1999). The MM system was treated with

the CHARMM36m force field (Klauda et al, 2010; Huang et al, 2017) for protein, lipid, and ions, and the CHARMM TIP3P model for waters, as in the classical MD simulations. Electrostatic interactions between the QM and MM subsystems were described using the Hamiltonian electrostatic coupling scheme (Laio et al, 2002; Olsen et al, 2019). The short-range electrostatic interactions between the QM partition and any MM atom within a cutoff of 30 a.u. from the QM region were explicitly considered, with a seventh-order multipole expansion of the QM electrostatic potential used for long-range interactions.

A Born–Oppenheimer molecular dynamics (BOMD) scheme was used for all the QM/MM simulations carried out here. Initially, all systems underwent geometry optimization through simulated annealing, with gradual reduction in the temperature of the QM/MM system by removing excess kinetic energy in each step (velocity multiplied by a factor of 0.95–0.99). Subsequently, systems were linearly reheated to 300 K and the target temperature increased using a Berendsen thermostat, with coupling strength of 5000 a.u. A timestep of 10 a.u. (~0.24 fs) was used both for the initial geometry optimization through annealing and for the heating protocol. Finally, systems with QM0, QM1 and QM3 partitions underwent 20 ps of QM/MM BOMD equilibration before running TI simulations. During the equilibration, a timestep of 20 a.u. (~0.48 fs) was employed and temperature was maintained constant by using a Nosé–Hoover thermostat (Nosé, 1984; Hoover, 1985) with a coupling frequency of 3500 cm$^{-1}$. Further details of the QM/MM BOMD simulations with the QM2 partition are provided in the section "QM/MM MD proton migration simulations".

## TI simulations of proton transfer

The free energy profiles of the proton transfers investigated here was determined through TI in its Blue Moon sampling implementation (Ciccotti and Ferrario, 2004; Ciccotti et al, 2005). The collective variables (CVs) used to describe proton transfer (PT) were based on the difference of distances between donor, proton and acceptor. Specifically, for the PT from E133[H] to D46[−], CV0 was defined as the difference between (i) the distance between the carboxylate oxygen atom of E133[H] and the proton initially bound to it and (ii) the same proton and the carboxylate oxygen atom of D46[−] initially hydrogen bonded to it. For the PT from E133[H] to its adjacent water molecule, CV1 was the difference between (i) the distance between the carboxylate oxygen atom and the transferred proton of E133[H] and (ii) the distance between the same proton and the oxygen atom of water. Similarly, the CV2 used to simulate proton transfer from D46[H] to E133 was the difference between (i) the distance between the carboxylate oxygen atom and the transferred proton of D46[H] and (ii) the distance between the same proton and the carboxylate oxygen atom of E133.

A total of 23, 17 and 15 independent constrained BOMD simulations were conducted for each TI calculation, respectively, with windows separated by 0.08 Å increments. At each CV value (either CV0 or CV1 or CV2), the systems were simulated for 2 ps and the last snapshot was taken to set up the next window. The constraint force at each CV value was calculated by averaging the Lagrange multiplier of the Shake algorithm after discarding the first 0.5 ps (i.e., when it reached convergence). The free energy profiles were then obtained by integrating the constraint force

along the CV using the trapezoid method. For all TI profiles, the kinetic correction term was found to negligible because fluctuations in the angle connecting donor, proton, and acceptor are minimal.

## QM/MM MD proton migration simulations

Starting with the final structure (CV1 = 0.35 Å) obtained from the TI proton transfer simulation, in which the proton was transferred from E133[H] to the adjacent water molecule, we ran a 40 ps QM/MM BOMD simulation with a timestep of 20 a.u. (~0.48 fs), maintaining the same constraint applied during TI. This served to sample an initial set of structures of the solvated hydronium ion forming an ion pair with negatively charged E133. We then applied a cluster analysis approach utilizing the *cluster* module of GROMACS. E133, D46, galactonate, and the first layer of surrounding water molecules (within 4 Å) were considered in the RMSD calculation, and clustering was performed with the *gromos* method (Daura et al, 1999) and a 0.04 Å cutoff. This approach resulted in the identification of nine distinct clusters. Subsequently, seven representative snapshots, corresponding to the centroids of the seven most populated clusters and covering over 95% of the analyzed trajectory, were selected as starting configurations for the subsequent QM/MM BOMD simulations to explore the migration of the excess proton. Notably, these structures are characterized by variations in their hydrogen bond networks and the number of connecting water molecules among E133, hydronium ion, and galactonate. Statistical information pertaining to the proton transfer dynamics of these QM/MM BOMD simulations is presented in Appendix Table S2.

## Quantification of electrostatics and water-occupancy calculations

Electrostatics was quantified with *g_elpot* (Kostritskii et al, 2021; Kostritskii and Machtens, 2023); source code, installation instructions, and usage recommendations can be found at https://jugit.fz-juelich.de/computational-neurophysiology/g_elpot. The distribution of electrostatic potential was calculated via the smooth particle mesh Ewald (SPME) method. For our system, the SPME potential was calculated on a grid of $256 \times 256 \times 208$ points with an inverse Gaussian width β of 20 nm$^{-1}$. The electrostatic potential near a specific residue (Fig. EV3F) was obtained by combining the electrostatic-potential in a sphere of 3 Å radius around the center of geometry of carboxyl groups of interest (OD1 and OD2 atoms of D46 and OE1 and OE2 atoms of E133) and water-occupancy time courses of the same region calculated by the *g_elpot* tool and averaging the potential over hydrated frames. To calculate the potential for the carboxyl groups of protonated D46 and E133 (Fig. 6B), we exploited the extended functionality of the tool (Kostritskii and Machtens, 2023). The electrostatic potential is an average of the SPME potential in a 0.15-nm sphere around the following atom groups (CHARMM naming convention): OD1, OD2, and CG of D46 and OE1, OE2, and CD of E133. Since the two selected (carboxylate) groups are chemically identical, the residue-specific short-range part of the potential is the same and, therefore, comparison of the potential values is justified. The time course of the electrostatic potential was averaged across each trajectory.

Water-occupancy maps were also generated by *g_elpot* and report on the fraction of frames with a node occupied by water.

## Markov state model construction

Markov state models (MSMs) were constructed from multiple unbiased MD trajectories using PyEMMA 2 software (Scherer et al, 2015). To better capture the movement of N- and C-terminal domains relative to each other, we selected every 5th Cα atom of the TM helices and measured the distances between each selected atom located on the N-terminal (TM1–6) and C-terminal (TM7–12) domains, respectively. This resulted in $27 \times 26 = 2809$ pairwise distances. To reduce the dimensionality, we used a time-lagged independent component analysis (tICA) with a lag time of 50 ns on a set of unbiased MD simulations to obtain the slowest collective motions. For a description of the major conformational changes, we kept the first two independent components (ICs) because they had slower timescales than the other ICs (Appendix Fig. S15b,c). Since the starting point of the unbiased simulations was the corresponding crystal structure (either inward- or outward-facing), only regions of constructed conformational space that are close to the starting conformations were sampled. To overcome the sampling gap between unbiased simulations that started with different crystal structures, we identified conformations that were closest to the central region of conformational space and, therefore, should represent different occluded states and initiated new unbiased simulations from them. This process was repeated iteratively until the implied timescales converged. A total of ~25 μs (double protonated substrate-bound DgoT) and ~37 μs (DgoT *apo*) simulation data were obtained and used for further data analysis, with individual trajectories of 400–800 ns in length (Appendix Fig. S15a). The implied timescales plot demonstrates the Markovian behavior after a lag time of ~80 ns (Appendix Fig. S15b,c). For constructing the MSM, a lag time of 100 ns was chosen from the implied timescales plot. For model validation, we performed a Chapman–Kolmogorov test with three metastable states for both substrate-bound and *apo* DgoT systems (Appendix Fig. S15d,e). Metastable states were identified using Perron-cluster cluster analysis (PCCA) (Deuflhard and Weber, 2005). For each system, the implied timescale plots (Appendix Fig. S15b,c) revealed two slow processes; therefore, we clustered the microstates into three metastable states.

## Data analysis and statistics

Each experimental dataset represented as means ± error bars was generated by performing the same experiment on multiple sensors (sensor numbers are given in the respective legends). Error bars represent standard deviations of the average value, and individual data points are given for $n < 5$. Hydration profiles (Figs. 3C and 4C) were calculated as the number of water molecules in 2 Å sections along the z-coordinate in each frame; average values and standard deviations are plotted. Violin plots were used to visualize the shape of data distribution and highlight trends. For statistical analysis of two groups (Fig. 8E), we used Mann–Whitney test, one-sided; significance is indicated as $*P < 0.05$.

## Data availability

Source data and galactonate parameters are provided at https://jugit.fz-juelich.de/n.dmitrieva/dgot-transport-mechanism/.

The source data of this paper are collected in the following database record: biostudies:S-SCDT-10_1038-S44318-024-00279-y.

## Peer review information

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

## Acknowledgements

We thank Drs Andre Bazzone, Bassam Haddad, Andrei Kostritskii, Piersilvio Longo and Jan-Philipp Machtens for helpful discussions and Meike Berndt for excellent technical support. This work was supported by the Deutsche Forschungsgemeinschaft (German Research Foundation) to ChF (FA 301/15–2), PC (CA 973/27-2) and MAP (AL 2511/1-2) as part of Research Unit FOR 2518, DynIon. The authors gratefully acknowledge computing time on the supercomputer JURECA at Forschungszentrum Jülich under grants dgoth and vglut-pt.

## Author contributions

**Natalia Dmitrieva**: Conceptualization; Data curation; Formal analysis; Validation; Investigation; Visualization; Writing—original draft; Writing—review and editing. **Samira Gholami**: Data curation; Formal analysis; Validation; Investigation; Visualization; Writing—review and editing. **Claudia Alleva**: Data curation; Formal analysis; Validation; Investigation; Visualization; Writing—review and editing. **Paolo Carloni**: Conceptualization; Supervision; Funding acquisition; Project administration; Writing—review and editing. **Mercedes Alfonso-Prieto**: Conceptualization; Supervision; Funding acquisition; Project administration; Writing—review and editing. **Christoph Fahlke**: Conceptualization; Supervision; Funding acquisition; Writing—original draft; Project administration; Writing—review and editing.

Source data underlying figure panels in this paper may have individual authorship assigned. Where available, figure panel/source data authorship is listed in the following database record: biostudies:S-SCDT-10_1038-S44318-024-00279-y.

## Funding

## Disclosure and competing interests statement

The authors declare no competing interests.

# Expanded View Figures

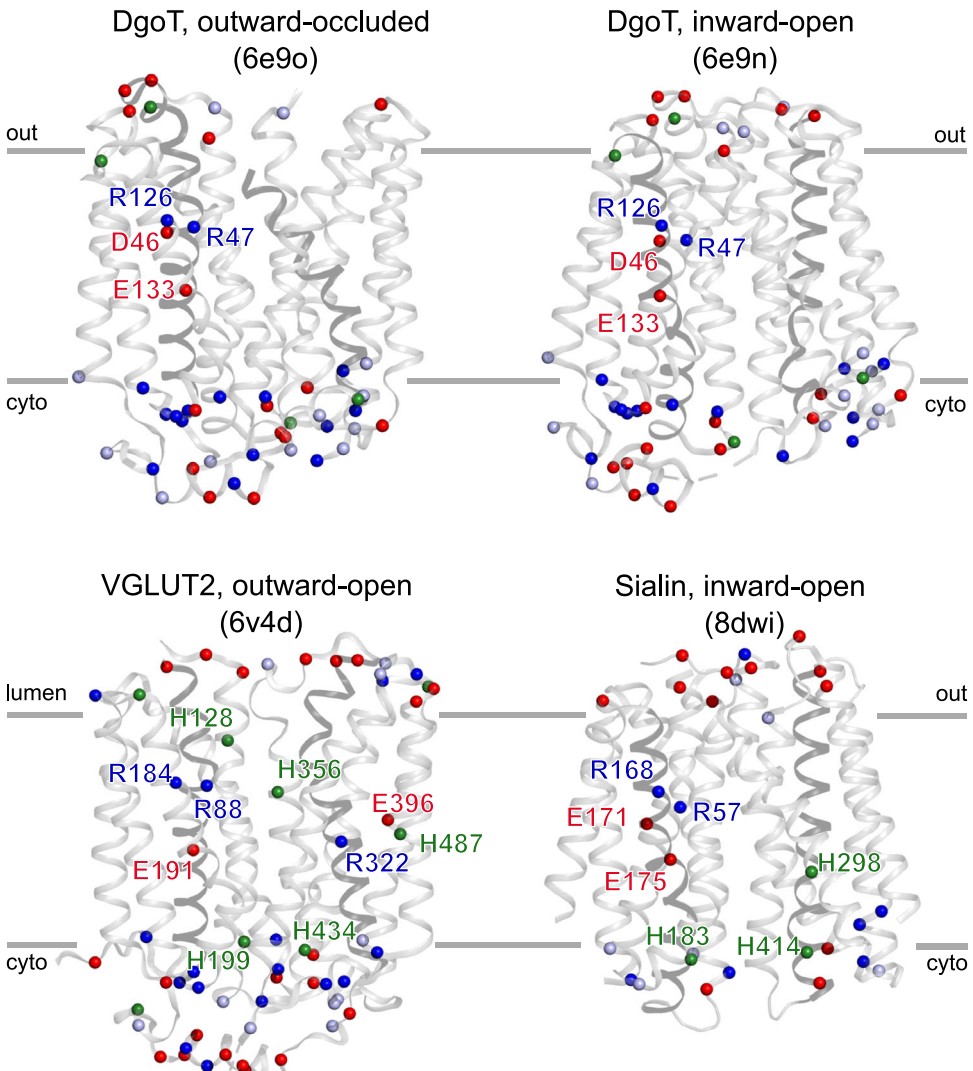

**Figure EV1.  Structural comparison of SLC17 members DgoT, VGLUT2 and sialin.**

PDB codes of the experimentally determined corresponding structures are shown between parentheses. The approximate location of the membrane is indicated with gray lines. Spheres represent the Cα position of Asp and Glu (red), His (green), Arg (dark blue) and Lys (light blue). Charged and titratable residues within the membrane are labeled.

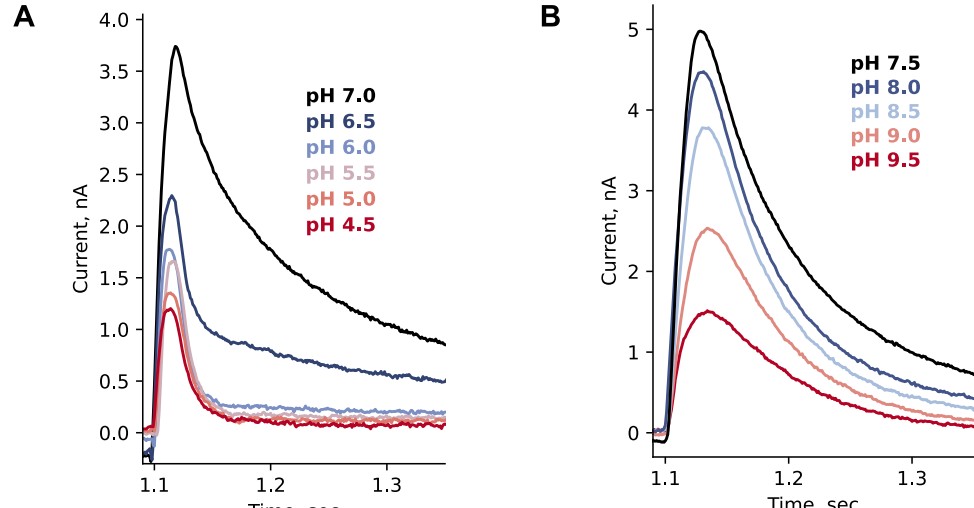

**Figure EV2.  pH dependence of WT DgoT currents measured by SSME upon application of 10 mM D-galactonate concentration jump.**

(A, B) Representative SSME currents obtained using the low time resolution set up (3 mm sensors) elicited by application of external solutions with various acidic (A) or alkaline (B) pH values.

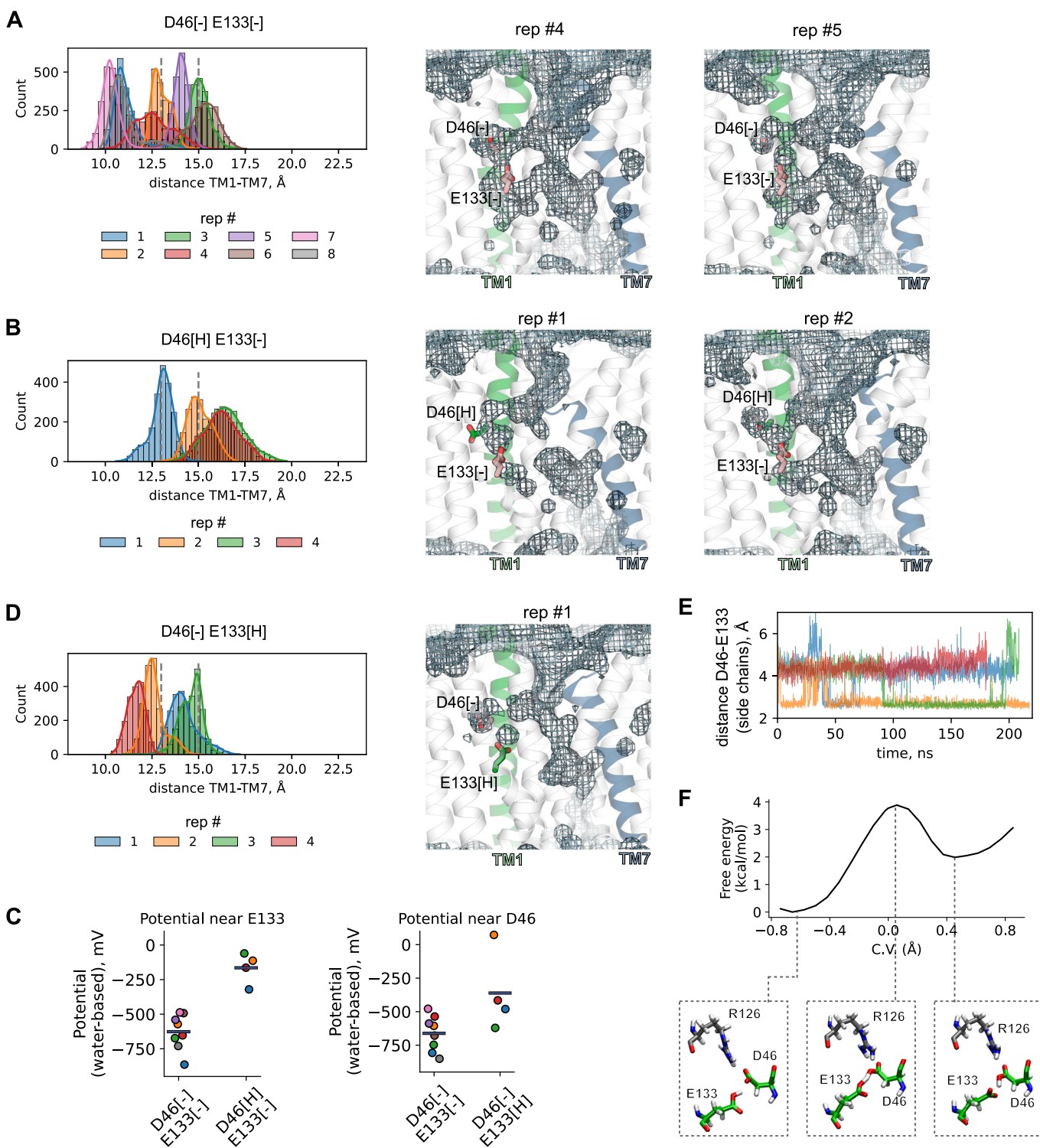

**Figure EV3.  Protonation of D46 and E133 in outward-facing *apo* DgoT.**

(**A–C**) Left: Probability densities for the extracellular gate opening in simulations with *apo* DgoT with deprotonated D46 and E133 ($n = 8$) (**A**), protonated D46 and deprotonated E133, $n = 4$ (**B**) or deprotonated D46 and protonated E133 ($n = 4$) (**C**). *Right*: Water occupancy map in selected replicas, contoured at an occupancy level of 0.2. (**D**) Time course of minimum distance between carboxyl groups of D46 and E133 in simulations with *apo* DgoT with deprotonated D46 and protonated E133. (**E**) Free energy profile for the proton transfer between D46 and E133, computed at the QM (BLYP)/MM level. The insets show representative starting, transition state, and final configurations. Error bars are omitted since they are smaller than the marker size. (**F**) Electrostatic potential near the carboxyl groups of deprotonated D46 (left) and E133 (right) in the unbiased MD simulations. Each data point is an average potential from a single trajectory.

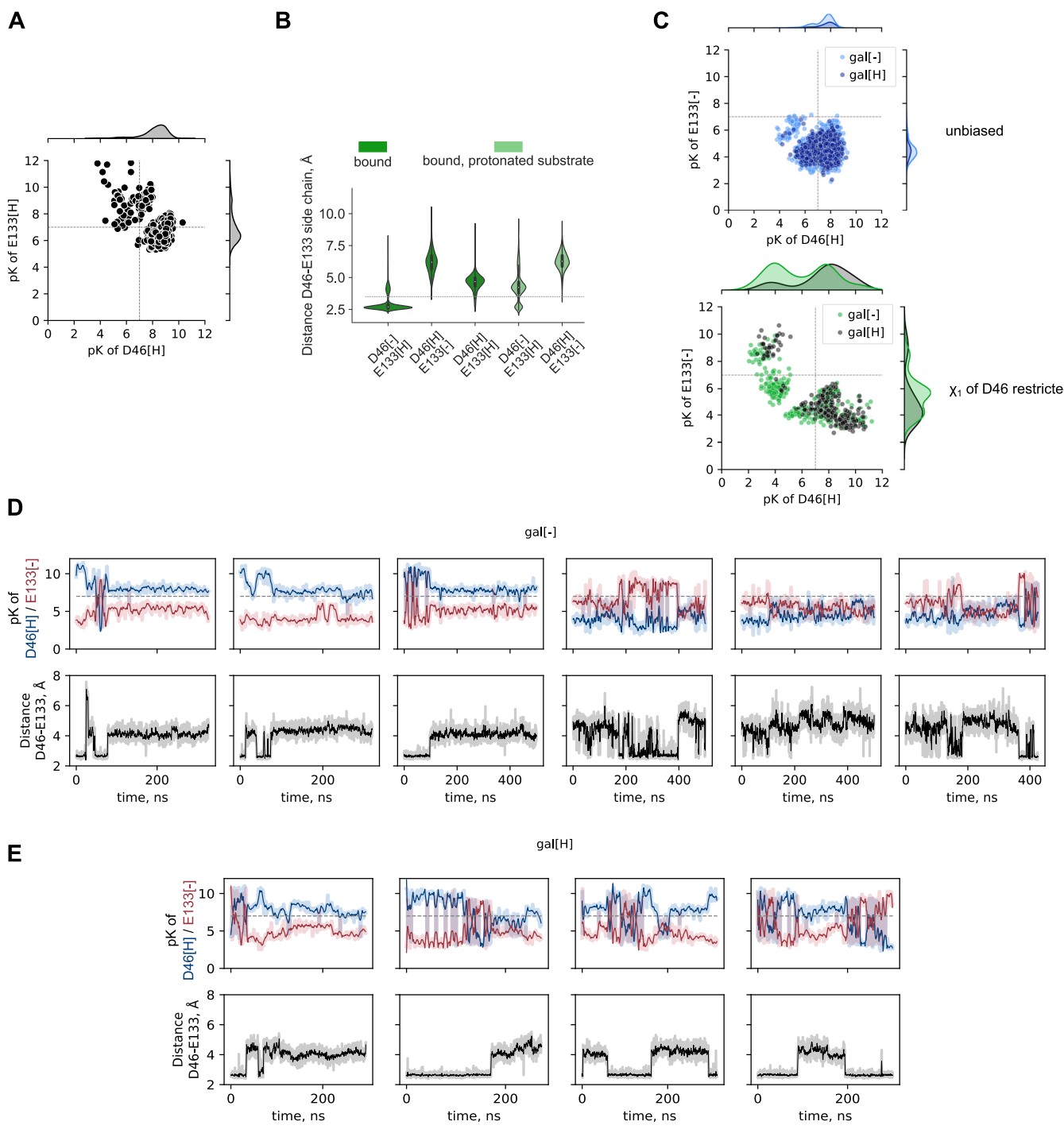

**Figure EV4.  pKₐ calculations with the PROPKA tool.**

(A) pK$_a$ of E133 versus pK$_a$ of D46 in simulations with inward-facing DgoT, D46 and E133 protonated and galactonate bound. (B) Probability densities for distances between side chains of D46 and E133 in simulations with inward-facing galactonate-bound DgoT (for simulations with deprotonated galactonate bound: $n = 10$ for simulations with only D46 or only E133 protonated, $n = 5$ for simulations with both D46 and E133 protonated; for simulations with protonated galactonate bound: $n = 4$). The central white dots represent the medians, thick black lines represent ranges between the 25th percentile (Q1) and the 75th percentile (Q3), the top and bottom points of the violin extend to the minimum and maximum values of the kernel density estimate (KDE). (C) pK$_a$ of E133 versus pK$_a$ of D46 in simulations with inward-facing DgoT with D46 protonated, E133 deprotonated and galactonate bound. *Top*: unbiased simulations, *bottom*: simulations with dihedral angle χ1 of D46 restrained at +70° (*close*-D46 conformation). (D, E) Time course of pK$_a$ of D46 and E133 and distance between side chains of D46 and E133 in simulations with inward-facing DgoT with D46 protonated, E133 deprotonated and χ1 of D46 restrained at +70° (*close*-D46). Galactonate was present in the binding site in its deprotonated (D) or its protonated (E) form.

