## [Peer Review File · The EMBO Journal]

Transport mechanism of DgoT, a bacterial homolog of SLC17 organic anion transporters

Christoph Fahlke, Natalia Dmitrieva, Samira Gholami, Claudia Alleva, Paolo Carloni, and Mercedes Alfonso-Prieto

Corresponding author(s): Christoph Fahlke (c.fahlke@fz-juelich.de)

Review Timeline:

Submission Date:	19th Mar 24
Editorial Decision:	22nd Apr 24
Revision Received:	2nd Sep 24
Editorial Decision:	27th Sep 24
Revision Received:	1st Oct 24
Accepted:	7th Oct 24

Editor: William Teale

Transaction Report:

Dear Prof. Fahlke,

Thank you again for the submission of your manuscript entitled "Transport mechanism of DgoT, a bacterial homolog of SLC17 organic anion transporters" (EMBOJ-2024-117337) and for your patience during the review process. We have now received the reports from three referees, which I copy below.

As you can see from their comments, while all referees ask for more detailed explorations into of certain aspects of the mechanism you propose, all of them point out the potential value of your study.

Based on the overall interest expressed in the reports, therefore, I would like to invite you to address the comments of all referees in a revised version of the manuscript. I should add that it is The EMBO Journal policy to allow only a single major round of revision and that it is therefore important to resolve the main concerns at this stage. I believe the concerns of the referees are reasonable and addressable, but please contact me if you have any questions, need further input on the referee comments or if you anticipate any problems in addressing any of their points. Many authors find a Zoom call with an editor a useful part of the process at this stage; I would be happy to meet you and discuss the referee reports once you have had a chance to digest them. If there are any points you would like to go through with me, please suggest a convenient time for a meeting.

Please, follow the instructions below when preparing your manuscript for resubmission.

I would also like to point out that as a matter of policy, competing manuscripts published during this period will not be taken into consideration in our assessment of the novelty presented by your study ("scooping" protection). We have extended this 'scooping protection policy' beyond the usual 3 month revision timeline to cover the period required for a full revision to address the essential experimental issues. Please contact me if you see a paper with related content published elsewhere to discuss the appropriate course of action.

Again, please contact me at any time during revision if you need any help or have further questions.

Thank you very much again for the opportunity to consider your work for publication. I look forward to your revision.

Best regards,

William

William Teale, Ph.D.
Editor
The EMBO Journal

When submitting your revised manuscript, please carefully review the instructions below and include the following items:

- 1) a .docx formatted version of the manuscript text (including legends for main figures, EV figures and tables). Please make sure that the changes are highlighted to be clearly visible.
- 2) individual production quality figure files as .eps, .tif, .jpg (one file per figure).
- 3) a .docx formatted letter INCLUDING the reviewers' reports and your detailed point-by-point response to their comments. As part of the EMBO Press transparent editorial process, the point-by-point response is part of the Review Process File (RPF), which will be published alongside your paper.
- 4) a complete author checklist, which you can download from our author guidelines ([https://wol-prod-cdn.literatumonline.com/pb-assets/embo-site/Author Checklist%20-%20EMBO%20J-1561436015657.xlsx](https://wol-prod-cdn.literatumonline.com/pb-assets/embo-site/Author%20Checklist%20-%20EMBO%20J-1561436015657.xlsx)). Please insert information in the checklist that is also reflected in the manuscript. The completed author checklist will also be part of the RPF.
- 5) Please note that all corresponding authors are required to supply an ORCID ID for their name upon submission of a revised manuscript.

6) We require a 'Data Availability' section after the Materials and Methods. Before submitting your revision, primary datasets produced in this study need to be deposited in an appropriate public database, and the accession numbers and database listed under 'Data Availability'. Please remember to provide a reviewer password if the datasets are not yet public (see <https://www.embopress.org/page/journal/14602075/authorguide#datadeposition>). If no data deposition in external databases is needed for this paper, please then state in this section: This study includes no data deposited in external repositories. Note that the Data Availability Section is restricted to new primary data that are part of this study.

Note - All links should resolve to a page where the data can be accessed.

8) For data quantification: please specify the name of the statistical test used to generate error bars and P values, the number (n) of independent experiments (specify technical or biological replicates) underlying each data point and the test used to calculate p-values in each figure legend. The figure legends should contain a basic description of n, P and the test applied. Graphs must include a description of the bars and the error bars (s.d., s.e.m.).

9) We would also encourage you to include the source data for figure panels that show essential data. Numerical data can be provided as individual .xls or .csv files (including a tab describing the data). For 'blots' or microscopy, uncropped images should be submitted (using a zip archive or a single pdf per main figure if multiple images need to be supplied for one panel). Additional information on source data and instruction on how to label the files are available at .

10) We replaced Supplementary Information with Expanded View (EV) Figures and Tables that are collapsible/expandable online (see examples in <https://www.embopress.org/doi/10.15252/embj.201695874>). A maximum of 5 EV Figures can be typeset. EV Figures should be cited as 'Figure EV1, Figure EV2" etc. in the text and their respective legends should be included in the main text after the legends of regular figures.

12) Our journal encourages inclusion of *data citations in the reference list* to directly cite datasets that were re-used and obtained from public databases. Data citations in the article text are distinct from normal bibliographical citations and should directly link to the database records from which the data can be accessed. In the main text, data citations are formatted as follows: "Data ref: Smith et al, 2001" or "Data ref: NCBI Sequence Read Archive PRJNA342805, 2017". In the Reference list, data citations must be labeled with "[DATASET]". A data reference must provide the database name, accession number/identifiers and a resolvable link to the landing page from which the data can be accessed at the end of the reference. Further instructions are available at .

Further instructions for preparing your revised manuscript:

We realize that it is difficult to revise to a specific deadline. In the interest of protecting the conceptual advance provided by the work, we recommend a revision within 3 months (21st Jul 2024). Please discuss the revision progress ahead of this time with the editor if you require more time to complete the revisions. Use the link below to submit your revision:

Referee #1:

The manuscript describes experimental and molecular simulation studies on a bacterial transporter, DgoT, which is an SLC17 homolog. Through biochemical and electrophysiology experiments, the authors characterized protonation changes during the transport of galactonate. The experiment determined the transport stoichiometry ratio of 2 protons : 1 galactonate in one transport cycle. The experiment also found that addition of galactonate to purified DgoT solution does not show any change in pH, suggesting that DgoT binds two protons prior to galactonate association and the substrate binding does not change the protonation states. Based on the experimental findings, computational molecular simulations with various methods propose a molecular model of the transport cycle which involves protonation/deprotonation of key acidic groups Asp46 and Glu133, movement of Arg47, opening/closures of channel gates, and conformational changes of alternating access of the entire protein structure. Further experiments with mutants were conducted and the mechanism of the transport cycle is discussed based on the computational model and the experiments with mutants.

The study reported is very elaborated and provides a comprehensive insight into the transport mechanism. I therefore believe that the manuscript is potentially suitable for publication. However, there are a couple of issues which may need to be clarified before publication. The authors may wish to address the issues described below.

1. I am slightly puzzled by the transport scheme proposed in Figure 9. The molecular simulations with MSM showed that the most stable conformation of the substrate bound state of DgoT is the inward-facing form (Figure 6C), which appears to be State 5 in Figure 9. However, the pH change experiment for the purified protein showed no change in pH upon the binding of the substrate, which contradicts to the scheme where release of a proton takes place from State 4 to 5. Alternatively, State 4 may be already the inward-facing form, although the image of State 4 looks to me like an outward-facing one. In this case, however, it does not seem to me very clear how the cycle proceeds. Despite that the driving force of the transport cycle should be the lower inner concentration of the substrate, i.e., release of the substrate in the inward-facing form, the proton release is proposed to occur without the substrate release and the conformational change of alternating access (State 4 to 5). No clear explanation on why the proton is released in the cycle without coupling with the release of the substrate and the alternating access conformational changes, while the purified protein sample in the equilibrium substrate concentration does not release a proton upon the substrate binding.

One possibility is that, given difficulty in accurate computational evaluation of energetics between the largely different protein conformations, the computational prediction of the populations of the conformation is erroneous and thus the inward-facing form is not the most stable one for the substrate bound state with the double protonation states. In this case, the proton release can be coupled with the alternating access conformational change leading to the substrate release in the transport cycle just as the

scheme in Figure 9, and the proton release does not take place in equilibrium of the purified sample where the substrate release is blocked.

The authors may wish to clarify and discuss this issue, i.e., what States 4 and 5 are and how the proton release takes place between those states without the release of the substrate, in light of their experimental findings and the computational analyses along with clear indication of what state corresponds to the substrate bound state in the equilibrium purified sample.

2. I could not find detailed information of MD trajectory lengths of unbiased MD simulations for different protonation state systems corresponding to Figures 2 and 4. Since large conformational changes seems to be involved upon changes of the protonation state and the substrate binding, long trajectory calculations of equilibration for relaxation of the conformations are critical to properly model those conformations. In addition, trajectory lengths for conformational samplings to calculate distributions such as Figures 2D,E and 4D are also necessary information to assess the computational validity, especially as the authors argue equilibrium shift based on the distribution.

Those information needs to be listed properly and validation of the equilibration, for example, showing time courses of RMSDs etc, is welcome.

Referee #2:

Recommendation: Publish in EMBO J. after revision

Comments:

The manuscript "Transport mechanism of DgoT, a bacterial homolog of SLC17 organic anion transporters" by Dmitrieva et al. combines electrophysiological experiments and computer simulations to investigate the working mechanism of a H⁺/galactonate co-transporter DgoT. The authors, for the first time, determined an H⁺:galactonate stoichiometry of 2 and revealed that DgoT takes 2 H⁺ before binding galactonate. Classical molecular dynamics (MD) simulations suggest that residues D46 and E133 are H⁺ acceptors, which is confirmed by electrophysiological characterization of mutants D46N and E133Q. Although the conformational transition from outward-facing (OF) to inward-facing (IF) is triggered by galactonate binding regardless of the protonation states of D46 and E133, deprotonation of D46 and E133 is required to complete the transport cycle. Regarding H⁺ and galactonate release, classical MD simulations suggest that H⁺ release from D46 is required for galactonate release. Free energy and quantum mechanics/molecular mechanics (QM/MM) calculations suggest that D46 likely releases its H⁺ to a deprotonated E133 or galactonate. Markov state modeling implies that the IF-to-OF transition is the rate-limiting step (RLS) in the transport cycle. An alternating-access mechanism was proposed. The study was comprehensive and conducted properly. The manuscript was well-organized and clearly written. It is appropriate for publication in EMBO J. However, I have a couple of concerns for the authors to address.

1. My primary concern lies in the timescales of H⁺ release mechanism in the IF state.

a) The authors claim that their QM/MM simulations suggest H⁺ transfer to galactonate or the solvent is viable over a sub-nanosecond timescale (page 10, lines 237-241). Though this may be true for the protonation of galactonate (Table S2), the H⁺ release from E133 when D46 stays protonated (a barrier of 8 kcal/mol, Fig. S5) and the H⁺ transfer from D46H to E133- (a barrier >10 kcal/mol, Fig. 5CD) are apparently much slower.

b) It appears that the H⁺ transfer between D46 and E133 is a two-step process. It should be described by 2-dimensional (2D) free energy sampling with two reaction coordinates (RCs): sidechain rotation of D46H (Fig. 5C) and H⁺ migration between D46H and E133- (Fig. 5D), while the authors computed free energies along each RC separately. Assuming the minimum free energy path in the 2D free energy surface is a simple superimposition of the two 1D free energy profiles, the barrier is about 16 kcal/mol, indicating the H⁺ transfer happens on the millisecond timescale.

c) I am glad that the authors recognize the possibility that the bound galactonate could be protonated, and the QM/MM simulations (Fig. S6) indicate that such a possibility cannot be ruled out at this stage. Assuming that D46H releases its H⁺ to the bound galactonate, will the protonated galactonate leave DgoT then deprotonate in the internal bulk, or will the protonated galactonate release the H⁺ to the internal bulk before leaving DgoT? Alternatively, could both scenarios be possible, pointing to a kinetic network model like those reported for the peptide transporter PepT (Li et al., *Biophys. J.* 2022, 121, 2266-2278) and the phosphate transporter PipT (Liu et al., *Biophys. J.* DOI: 10.1016/j.bpj.2024.03.035)? Can experiments clarify this, for instance, using transportable analogs of protonated galactonate in H⁺ flux measurements?

2. The authors built Markov State Models to assess the relative stabilities of various conformational states (Fig. 6). While it makes sense that galactonate binding stabilizes the D46H/E133H-IF state and the D46-/E133--OF state is stable without a bound galactonate, the authors do not report transition probabilities between these conformational states. Cannot Markov State Modeling estimate these? I think the transition probabilities would be helpful in determining RLSs in the transport cycle. I agree that H⁺ transfer is less likely to be an RLS, as is the case in other MFS proteins like PepT (Minhas et al., *eLife* 2018, 7, e34995; Li et al., *Biophys. J.* 2022, 121, 2266-2278). However, I am afraid it is not clear at this stage whether the OF-to-IF or the IF-to-OF is faster. Have scientists measured the turnover rate of DgoT? Or am I missing anything here?

Having said these, I believe the current study offers valuable insights into the functioning cycle of DgoT, but the H⁺ release mechanism remains largely elusive. Further studies employing faster reactive methods like multistate reactive MD and likely kinetic modeling may help resolve these uncertainties. So, it would be better that the authors revise as suggested above, temper their tone a bit, and state their limitations regarding the current study.

Referee #3:

The manuscript by Dmitrieva et al describe the molecular basis and mechanism of DgoT, a galactonate transporter, by a range of experimental and computational methods. The work is solid and I have some major and minor points, which authors must answer before the manuscript can be published.

1. What is the driving force for protons that protonate D46 and E133? What is the origin of these protons? It is also difficult to envisage loading two proton sites in protein matrix, one after another. Loading of the second proton must be even more energy demanding given the likely cooperativity between the two acidic residues in vicinity to each other. Authors must probe this point to strengthen their model of two proton loading in state 1 and 2 of their mechanistic model, which resembles the one presented by Leano et al. Also, it is unclear how the loading of two protons and galactonate to its binding site will be affected by proton electrochemical gradient across the membrane. Authors can provide semi-quantitative arguments in the favor of their charge transfer events.
2. Lines 143-149 - it is unclear how many binding events of galactonate to protein occurred? Fig. 3D gives result from a single replica. It would be important to strengthen the notion of galactonate binding in a given charge state by more simulation sampling and alternative simulation approaches. For instance, they can also probe effect of charge on unbinding of (bound) substrate.
3. Lines 151-157 - The analysis presented in this paragraph will benefit if authors provide hydration/Dehydration effects when charge states of D46/E133 changes. Conformational changes coupled to hydration effects are well-known.
4. Lines 188-onwards - authors must show the data also in MD simulation time series fashion. Probability density plots are indeed good to reveal the overall occupancies, but for instance when and how frequent galactonate left the pocket would be important to show and discuss in more detail. Also, it is unclear what is the main driving force for substrate release? Is it electrostatic repulsion by anionic D46, hydration changes and/or coupled conformational changes.
5. Proton release from inward facing DgoT - it is often extremely challenging to track the sequence of events. It would benefit to the reader, if authors discuss alternative scenarios. Moreover, pKa estimates from PB-based approaches or even empirical propka-like approaches would be good to obtain. Also on simulation snapshots to show how pKa changes upon conformational changes. See a recent work on highlighting extent of conformational changes in standard and non-standard charge states of a protein - <https://doi.org/10.1021/acs.jpcc.3c07421>
6. There is no figure in main text, like an introductory figure, showing the protein architecture, membrane around it and the location of the residues with respect to membrane. I think this should be provided for readers to get acquainted with the topic early on.
7. What do the dotted lines represent in Fig. 2d, 2e, etc?
8. E133Q structure was used to construct WT system. Authors must provide more details on how they relaxed the structure after converting mutant to WT.
9. Galactonate charge/parameters must be provided in supplementary information for reproducibility.
10. MD simulation methods - pKa calculations can be used to decipher ground state protonation states. Also, provide information if histidines were delta or epsilon protonated in their charge neutral states.
11. Missing residues were modeled - how was the protein relaxed after modeling - can these affect authors results and conclusions?
12. The time step in QM/MM equilibration steps was small and also in production runs. Why is that? With larger time step, were there problems in simulations?

William Teale, Ph.D.
Editor
The EMBO Journal

RE: Re-Submission of Revised Manuscript EMBOJ-2024-117337

Dear William,

we would like to submit a revised version of our manuscript entitled *Transport mechanism of DgoT, a bacterial homolog of SLC17 organic anion transporters* by Natalia Dmitrieva, Samira Gholami, Claudia Alleva, Paolo Carloni, Mercedes Alfonso-Prieto and Christoph Fahlke to be considered for publication in the *EMBO Journal*. The referees provided clear and constructive criticisms that we have fully addressed in the following text and in the revised manuscript. Changes in the manuscript are marked as red, and specific responses to the referees' comments and other manuscript changes are outlined below.

As per the request of the editorial office, we have also deposited the source data in a publicly available repository (<https://jugit.fz-juelich.de/n.dmitrieva/dgot-transport-mechanism/>) and amended accordingly the Data Availability section. We have also replaced the Supporting Information by Expanded View Figures and an Appendix.

Reviewers' Comments:

Reviewer #1: *The manuscript describes experimental and molecular simulation studies on a bacterial transporter, DgoT, which is an SLC17 homolog. Through biochemical and electrophysiology experiments, the authors characterized protonation changes during the transport of galactonate. The experiment determined the transport stoichiometry ratio of 2 protons : 1 galactonate in one transport cycle. The experiment also found that addition of galactonate to purified DgoT solution does not show any change in pH, suggesting that DgoT binds two protons prior to galactonate association and the substrate binding does not change the protonation states. Based on the experimental findings, computational molecular simulations with various methods propose a molecular model of the transport cycle which involves protonation/deprotonation of key acidic groups Asp46 and Glu133, movement of Arg47, opening/closures of channel gates, and conformational changes of alternating access of the entire protein structure. Further experiments with mutants were conducted and the mechanism of the transport cycle is discussed based on the computational model and the experiments with mutants.*

The study reported is very elaborated and provides a comprehensive insight into the transport mechanism. I therefore believe that the manuscript is potentially suitable for publication.

However, there are a couple of issues which may need to be clarified before publication. The authors may wish to address the issues described below.

“1. I am slightly puzzled by the transport scheme proposed in Figure 9. The molecular simulations with MSM showed that the most stable conformation of the substrate bound state of DgoT is the inward-facing form (Figure 6C), which appears to be State 5 in Figure 9. However, the pH change experiment for the purified protein showed no change in pH upon the binding of the substrate, which contradicts to the scheme where release of a proton takes place from State 4 to 5. Alternatively, State 4 may be already the inward-facing form, although the image of State 4 looks to me like an outward-facing one. In this case, however, it does not seem to me very clear how the cycle proceeds. Despite that the driving force of the transport cycle should be the lower inner concentration of the substrate, i.e., release of the substrate in the inward-facing form, the proton release is proposed to occur without the substrate release and the conformational change of alternating access (State 4 to 5). No clear explanation on why the proton is released in the cycle without coupling with the release of the substrate and the alternating access conformational changes, while the purified protein sample in the equilibrium substrate concentration does not release a proton upon the substrate binding.

One possibility is that, given difficulty in accurate computational evaluation of energetics between the largely different protein conformations, the computational prediction of the populations of the conformation is erroneous and thus the inward-facing form is not the most stable one for the substrate bound state with the double protonation states. In this case, the proton release can be coupled with the alternating access conformational change leading to the substrate release in the transport cycle just as the scheme in Figure 9, and the proton release does not take place in equilibrium of the purified sample where the substrate release is blocked.

The authors may wish to clarify and discuss this issue, i.e., what States 4 and 5 are and how the proton release takes place between those states without the release of the substrate, in light of their experimental findings and the computational analyses along with clear indication of what state corresponds to the substrate bound state in the equilibrium purified sample.”

In the pH change experiment with purified DgoT, the transporter is in its *apo* state before galactonate application, and, based on the results of MSM, predominantly in the outward-facing conformation. We therefore assume that our experiment captures only substrate binding steps (i.e. some of the states between 2 to 5). If H⁺ binding were the consequence of galactonate association, we would expect the measured pH to increase upon galactonate association. This was not observed, and we conclude that transporter protonation precedes substrate association.

We do not have a straightforward way to monitor the starting and/or the final protein conformation in this experiment. However, even if a significant fraction of DgoT existed in the inward-facing conformation, this conclusion - that transporter protonation precedes substrate association - would be the same.

If the transport cycle normally proceeded after galactonate binding, DgoT would translocate to the inward-facing conformation and subsequently release H⁺. Since DgoT in detergent micelles is not exposed to electrical or chemical gradients, such progression within the transport cycle is therefore unlikely or at least very slow. In the past, the pH change

experiment with purified proteins was applied to EmrE (Soskine et al. (2004). *J Biol Chem* **279**, 9951-9955) and LacY (Smirnova et al. (2012). *Proc Natl Acad Sci USA* **109**, 16835-16840) transporters, and similarly interpreted. We are discussing possible limitations in interpreting these experiments on lines 461-471.

In the transport cycle model, state 4 represents the fully occluded conformation, in which the binding site is not accessible from either membrane side. In the original version, the structural differences from states 3 or 5 are indeed difficult to recognize, since the gating helices change position only slightly in these three states. We updated the scheme to make it clearer and more coherent with the text; please see the revised version of Figure 10. The conformation representing state 4 was replaced with a more suitable snapshot, in which the protein is more symmetrical. The proton and substrate dissociation events are now indicated between states 5 to 8, instead of 4 to 7. We have also clarified that there might exist mechanisms of substrate and proton release other than the one presented in the scheme, e.g. dissociation of H⁺ and substrate together as protonated galactonate; see lines 299-301 in section “Proton release from inward-facing DgoT”, as well as lines 430-434, 437-443 and 511-520 in the Discussion.

“2. I could not find detailed information of MD trajectory lengths of unbiased MD simulations for different protonation state systems corresponding to Figures 2 and 4. Since large conformational changes seems to be involved upon changes of the protonation state and the substrate binding, long trajectory calculations of equilibration for relaxation of the conformations are critical to properly model those conformations. In addition, trajectory lengths for conformational samplings to calculate distributions such as Figures 2D,E and 4D are also necessary information to assess the computational validity, especially as the authors argue equilibrium shift based on the distribution. Those information needs to be listed properly and validation of the equilibration, for example, showing time courses of RMSDs etc, is welcome. “

In the revised version of the Appendix (formerly Supplementary Information), we added a summary table (Appendix Table S3) with all information about classical MD simulations and figures (Appendix Figures S3A-B and S7) with independent analysis of trajectories used for Figures 3D, E and 5D (please note the updated numbering of the main text figures). The overlap of probability distributions for 10-ns blocks with the final profile was then analyzed to confirm that distances between gating helices were reasonably converged in those

trajectories (new Appendix Figure S3C). This information was also added to the Methods section (lines 739-742).

During this analysis we realized that three replicas for *apo* DgoT with deprotonated D46 and E133 were longer (800 ns) than the rest; they had been produced to collect data for the MSM construction. To improve comparability, we decided to use only the first 200 ns for all conditions and updated Figure 3 accordingly.

Reviewer #2: *The manuscript "Transport mechanism of DgoT, a bacterial homolog of SLC17 organic anion transporters" by Dmitrieva et al. combines electrophysiological experiments and computer simulations to investigate the working mechanism of a H⁺/galactonate co-transporter DgoT. The authors, for the first time, determined an H⁺:galactonate stoichiometry of 2 and revealed that DgoT takes 2 H⁺ before binding galactonate. Classical molecular dynamics (MD) simulations suggest that residues D46 and E133 are H⁺ acceptors, which is confirmed by electrophysiological characterization of mutants D46N and E133Q. Although the conformational transition from outward-facing (OF) to inward-facing (IF) is triggered by galactonate binding regardless of the protonation states of D46 and E133, deprotonation of D46 and E133 is required to complete the transport cycle. Regarding H⁺ and galactonate release, classical MD simulations suggest that H⁺ release from D46 is required for galactonate release. Free energy and quantum mechanics/molecular mechanics (QM/MM) calculations suggest that D46 likely releases its H⁺ to a deprotonated E133 or galactonate. Markov state modeling implies that the IF-to-OF transition is the rate-limiting step (RLS) in the transport cycle. An alternating-access mechanism was proposed. The study was comprehensive and conducted properly. The manuscript was well-organized and clearly written. It is appropriate for publication in EMBO J. However, I have a couple of concerns for the authors to address.*

1.a) My primary concern lies in the timescales of H⁺ release mechanism in the IF state. The authors claim that their QM/MM simulations suggest H⁺ transfer to galactonate or the solvent is viable over a sub-nanosecond timescale (page 10, lines 237-241). Though this may be true for the protonation of galactonate (Table S2), the H⁺ release from E133 when D46 stays protonated (a barrier of 8 kcal/mol, Fig. S5) and the H⁺ transfer from D46H to E133- (a barrier >10 kcal/mol, Fig. 5CD) are apparently much slower.

We thank the reviewer for pointing this out. Indeed, the sub-nanosecond timescale mentioned in the original version of the manuscript referred only to the proton transfer to galactonate or the solvent. The other processes involved in the D46 deprotonation pathway proposed here (D46 sidechain rotation, E133 deprotonation and proton transfer from D46 to E133) have larger barriers (10, 8 and 4 kcal/mol, respectively), which make these steps much slower. To avoid any misunderstandings, we have now rewritten the corresponding sentence (lines 286-287) and have emphasized in the Discussion that Markov state modeling suggests the reorientation of the empty transporter to be the rate-limiting step of the transport cycle (lines 445-446). We have also mentioned the limitations of the energetics computed based on 1D free energy profiles and simple collective variables (lines 270-272 and 295-301).

1. b) It appears that the H⁺ transfer between D46 and E133 is a two-step process. It should be described by 2-dimensional (2D) free energy sampling with two reaction coordinates (RCs): sidechain rotation of D46H (Fig. 5C) and H⁺ migration between D46H and E133- (Fig. 5D), while the authors computed free energies along each RC separately. Assuming the minimum free energy path in the 2D free energy surface is a simple superimposition of the two 1D free energy profiles, the barrier is about 16 kcal/mol, indicating the H⁺ transfer happens on the millisecond timescale.

We agree with the reviewer that the barriers calculated here on the basis of 1D free energy profiles are not accurate enough to estimate the timescale of proton release from D46 in the IF state. As indicated in the previous reply, we have now mentioned this point on lines 270-272 and 295-301. We have also discussed the need for further investigation of this (and other) possible proton release pathway(s) using 2D free energy sampling and collective variables optimized to describe proton transport processes (lines 430-434).

Nonetheless, we would like to clarify that the mechanism proposed here for proton release from D46 involves three steps, i.e. D46 sidechain rotation, E133 deprotonation and proton transfer from D46 and E133. The classical MD simulations presented in the original version of the manuscript, varying the protonation state of both acidic residues, indicated that rotation of D46 to the *close* conformation is needed for proton transfer between the two acidic residues to occur (Figure 6A-B). When the sidechain of D46[H] is in either the far or the intermediate conformations (Figure 6C), the distance with E133[-] is longer and there are no water molecules bridging the two acidic residues, preventing effective proton transfer. As suggested by reviewer 3, we have now additionally included the results of PROPKA calculations (lines 243-255) showing that, only when D46 is in the *close* conformation its pKa is lower than that of E133, thus favoring proton transfer. Therefore, we surmise that the D46 sidechain rotation might occur before the two proton transfer steps and thus its energetics can be studied separately, as done here.

1.c) I am glad that the authors recognize the possibility that the bound galactonate could be protonated, and the QM/MM simulations (Fig. S6) indicate that such a possibility cannot be ruled out at this stage. Assuming that D46H releases its H⁺ to the bound galactonate, will the protonated galactonate leave DgoT then deprotonate in the internal bulk, or will the protonated galactonate release the H⁺ to the internal bulk before leaving DgoT? Alternatively, could both scenarios be possible, pointing to a kinetic network model like those reported for the peptide transporter PepT (Li et al., Biophys. J. 2022, 121, 2266-2278) and the phosphate transporter PipT (Liu et al., Biophys. J. DOI: 10.1016/j.bpj.2024.03.035)? Can experiments clarify this, for instance, using transportable analogs of protonated galactonate in H⁺ flux measurements?

We thank the reviewer for this comment. As presented in the original version of the manuscript, our QM/MM MD simulations indicate that the proton can be released either

bound to galactonate or directly to the intracellular solution through water molecules (Appendix Table S2). Following the reviewer's suggestion, we now additionally mention that protonated galactonate could reach the internal bulk as such and then release the proton to the intracellular solution (as the pKa of galactonate in water is 3.39) or deprotonate while it is leaving DgoT (lines 437-443). Although we cannot predict which of the two scenarios is more favorable, it is unlikely that galactonate release and/or deprotonation are the rate-limiting step of the proton/substrate release mechanism, as dissociation was observed in unbiased classical MD simulations for both protonated and deprotonated galactonate (Appendix Table S1). In addition, we have suggested the need for kinetic network modeling to investigate both scenarios, as well as compared with other proton-coupled transporters in which the functional cycle includes substrate protonation (lines 511-520).

We have not come up with a strategy to experimentally address these open points. In SSME experiments, we usually observe multiple electrogenic reactions simultaneously (i.e. substrate binding and the following conformational changes). The reaction corresponding to the conformational change can be isolated easily by pre-incubation of the protein with the substrate; however, the isolation of the substrate binding or release reaction steps is not straightforward. Additionally, the release of the protonated galactonate is most likely not electrogenic, therefore it would be undetectable in our experimental setup.

We agree that it would be interesting to test such analogs of protonated D-galactonate experimentally; however, we did not find any commercially available compound, except for L-galactonamide (PubChem CID 192911). However, DgoT appears to be stereo-selective, as the closely related L-galactonate has been shown experimentally to be unable to bind or be transported by DgoT (<https://ediss.sub.uni-hamburg.de/handle/ediss/8786>). We therefore expect that L-galactonamide cannot act as transportable analog of protonated D-galactonate.

2. The authors built Markov State Models to assess the relative stabilities of various conformational states (Fig. 6). While it makes sense that galactonate binding stabilizes the D46H/E133H-IF state and the D46-/E133--OF state is stable without a bound galactonate, the authors do not report transition probabilities between these conformational states. Cannot Markov State Modeling estimate these? I think the transition probabilities would be helpful in determining RLSs in the transport cycle. I agree that H⁺ transfer is less likely to be an RLS, as is the case in other MFS proteins like PepT (Minhas et al., eLife 2018, 7, e34995; Li et al., Biophys. J. 2022, 121, 2266-2278). However, I am afraid it is not clear at this stage whether the OF-to-IF or the IF-to-OF is faster. Have scientists measured the turnover rate of DgoT? Or am I missing anything here?

To characterize the transition probabilities in our Markov models, we estimated mean first passage times (MFPT) between the metastable states. For both *apo* and galactonate-bound systems, MFPT between the OF and IF states revealed that the major conformational changes have a dominant direction depending on substrate occupation. As shown in the figure below, outward-to-inward transition of the galactonate-bound transporter (~16 μ s in panel A) is faster

than the inward-to-outward transition of the *apo* protein ($\sim 20\text{-}30\ \mu\text{s}$, B), leaving the reorientation of the empty transporter the slowest conformational change in the cycle.

SSME experiments provide an estimate of the rates of conformational changes from current decay time. For WT DgoT at low pH or E133Q DgoT (i.e. under non-transport conditions), the pre-steady state reactions correspond to galactonate-induced conformational changes with observed rates up to $100\text{-}120\ \text{s}^{-1}$ or transition times of $1/k_{\text{obs}} \sim 8\text{-}10\ \text{ms}$ at saturating galactonate concentration (Fig. 8E and Table 1). Although experimental rates might be underestimated due to limitations of the method, they are still orders of magnitude slower than rates of the outward-to-inward transition predicted by MSM. Transition rates predicted by Markov models are more often erroneous, since they do not describe equilibrium properties (Bowman, G.R., Pande, V.S., Noé, F. (Eds.), 2014. *An Introduction to Markov State Models and Their Application to Long Timescale Molecular Simulation*, Advances in Experimental Medicine and Biology. Springer Netherlands, Dordrecht). We therefore decided against adding the information on mean first passage times to the manuscript. Instead, we chose to keep using stationary probability distributions, rather than MFPT, to demonstrate the directionality of the major conformational changes and illustrate the differences between the systems with and without galactonate bound.

As already given in the earlier manuscript version we state on lines 333-338: *The outward-open state was not adopted by substrate-bound DgoT: it was only adopted by apo DgoT, for which it represents the state with highest stationary probability (Fig. 7D). Given that substrate binding and release are fast processes compared with the conformational changes, our data show that reorientation of the empty transporter upon substrate release is the rate-limiting step in the transport cycle.*

3. Having said these, I believe the current study offers valuable insights into the functioning cycle of DgoT, but the H⁺ release mechanism remains largely elusive. Further studies employing faster reactive methods like multistate reactive MD and likely kinetic modeling may help resolve these uncertainties. So, it would be better that the authors revise as suggested above, temper their tone a bit, and state their limitations regarding the current study.

As mentioned above, we have modified the section “Proton release from inward-facing DgoT” to describe the limitations of our multiscale simulations and point out the possibility of proton release pathways other than the one studied here (lines 295-301). We have also

expanded the discussion of the limitations of our study in the Discussion (lines 430-434). In addition, we have mentioned as outlook in the Discussion that other computational methods, such as adaptive QM/MM, multistate reactive MD and kinetic network modeling, would be needed to unravel the proton release pathways (lines 430-434 and lines 511-520).

Reviewer #3: *The manuscript by Dmitrieva et al describe the molecular basis and mechanism of DgoT, a galactonate transporter, by a range of experimental and computational methods. The work is solid and I have some major and minor points, which authors must answer before the manuscript can be published.*

1.a. What is the driving force for protons that protonate D46 and E133? What is the origin of these protons?

In simulations, as well as in the majority of the experiments, DgoT was tested without H⁺ gradient. The natural habitat of *E. coli* is the mammalian intestine, where it is exposed to neutral and alkaline pH (small intestine) or to acidic pH between pH 5.5 to 6.5 (colon) (Yamamura et al. (2023). *Frontiers in Microbiomes* **2**, 1192316, doi:10.3389/frmbi.2023.1192316). Acidic pHs are generated by the fermentation of dietary fiber and saccharides by intestinal bacteria. At external acidic pH, a higher intracellular pH generates an inwardly directed H⁺ gradient that provides a driving force for galactonate uptake (Slonczewski, et al. (1981). *Proc Natl Acad Sci USA* **78**, 6271–6275, doi:10.1073/pnas.78.10.6271.) We are now providing this information on lines 521-525.

1.b. It is also difficult to envisage loading two proton sites in protein matrix, one after another. Loading of the second proton must be even more energy demanding given the likely cooperativity between the two acidic residues in vicinity to each other. Authors must probe this point to strengthen their model of two proton loading in state 1 and 2 of their mechanistic model, which resembles the one presented by Leano et al. Also, it is unclear how the loading of two protons and galactonate to its binding site will be affected by proton electrochemical gradient across the membrane. Authors can provide semi-quantitative arguments in the favor of their charge transfer events.

For H⁺-coupled inward transport of galactonate, the transporter needs to be transiently protonated from the external medium and release H⁺ to the internal medium. Protonation of D46 or E133 requires water accessibility; it is promoted by low external pH and negative electrostatic potential. To further address the concern of the reviewer, we have now analyzed the distribution of water molecules in our simulations with *apo* protein and proposed a protonation mechanism of the two acidic residues in outward-facing, *apo* DgoT, based on this analysis and QM/MM simulations. These new results are shown in Fig. EV3 and discussed on lines 145-166. In short, when both acidic residues are deprotonated, they are connected to the extracellular solution via water molecules, and protonation can take place via a Grotthus

mechanism. If D46 is protonated first, E133 will remain water accessible from the outside and thus protonatable. Protonation of E133 sequesters the unprotonated D46 from the extracellular solution, but results in the formation of a hydrogen bond between D46[-] and E133[H]; thus, the latter can act as a proton relay to protonate D46.

1.c. Also, it is unclear how the loading of two protons and galactonate to its binding site will be affected by proton electrochemical gradient across the membrane.

The strict 2 H⁺ - 1 galactonate coupling, together with the sole presence of two protonation sites (D46 and E133) coupled to galactonate transport, indicates that there are no additional proton pathways in DgoT. In its outward-facing conformation, DgoT is only protonated from the external solution, and changes in intracellular pH can not affect protonation and galactonate binding. The alternating access mechanism described in the manuscript and summarized in Figure 10 results in independence of transporter loading in one conformation from the pH on the other membrane side, i.e. proton electrochemical gradient. The gradient, however, will affect transport, since it affects the likelihood of translocation events in the opposite direction.

2.a. Lines 143-149 - it is unclear how many binding events of galactonate to protein occurred? Fig. 3D gives results from a single replica. It would be important to strengthen the notion of galactonate binding in a given charge state by more simulation sampling and alternative simulation approaches.

Out of the 15 simulations run (i.e. 5 replicas for each of the three protonation state combinations of D46 and E133, see new Appendix Table S3), we observed a total of three galactonate binding events, one in the system with DgoT with only E133 protonated and two in the system with doubly protonated DgoT. We are now showing time courses of extracellular gate openings (distance TM1-TM7) and distances between binding residue R47 and closest galactonate molecule for all replicas in the new Appendix Figure S4.

Galactonate in solution is predicted to have a pKa of 3.39 in bulk water and therefore should be deprotonated under physiologically relevant pH. Thus, we considered that it binds to DgoT in deprotonated form. We added this information on lines 168-169. As mentioned above, we tested three charge combinations of D46 and E133, with either the former, or the latter or both protonated (see new Appendix Table S3).

Since we observed multiple spontaneous binding events in unbiased MD simulations, we did not see the need to use enhanced sampling techniques to further probe galactonate binding.

2. b. For instance, they can also probe effect of charge on unbinding of (bound) substrate.

Galactonate binding to outward-facing conformation triggers the transition of DgoT to the occluded state with extracellular gate closed (as illustrated in Figure 3C, E), which prevents substrate release back to the extracellular solution. This is mentioned on lines 181-182.

3. Lines 151-157 - The analysis presented in this paragraph will benefit if authors provide hydration/dehydration effects when charge states of D46/E133 changes. Conformational changes coupled to hydration effects are well-known.

To address this question, we analyzed changes in the number of water molecules in the neck region as a function of the gate opening for different protonation states of D46 and E133 (new Appendix Figure S5). As expected, the protonation state determines the gate opening, but leaves the number of water molecules at a given gate opening unchanged. This indicates that only the gate opening, but not the charge state, determines the number of water molecules in the neck. We now provide this information on lines 197-199.

4. Lines 188-onwards - authors must show the data also in MD simulation time series fashion. Probability density plots are indeed good to reveal the overall occupancies, but for instance when and how frequent galactonate left the pocket would be important to show and discuss in more detail. Also, it is unclear what is the main driving force for substrate release? Is it electrostatic repulsion by anionic D46, hydration changes and/or coupled conformational changes.

We added to the new Appendix Fig S6 time courses of the intracellular gate opening (distance F137-W373) and distance between R47 and the closest galactonate atom (as proxy for substrate dissociation) for all replicas in systems with D46 deprotonated (i.e. 18 simulations). In most cases, the substrate was released within the first 50 ns of the production run. We believe that this behavior is related to the fact that all simulations started with inward-facing crystal structure, in which the intracellular gate was open.

The driving force for the substrate release is not immediately clear. Since galactonate does not interact directly with D46 or E133, we hypothesize that the effect of their protonation state on substrate release is more complex than simple electrostatic repulsion. Moreover, E133 is closer to the binding site than D46, but its deprotonation does not seem to induce substrate release. Thus, we concluded that the protonation state of D46 and E133 is coupled to conformational changes in the protein, which regulate substrate release. This information is given on lines 223-230.

5. Proton release from inward facing DgoT - it is often extremely challenging to track the sequence of events. It would benefit to the reader, if authors discuss alternative scenarios.

Moreover, pKa estimates from PB-based approaches or even empirical propka-like approaches would be good to obtain. Also on simulation snapshots to show how pKa changes upon conformational changes. See a recent work on highlighting extent of conformational changes in standard and non-standard charge states of a protein - <https://doi.org/10.1021/acs.jpcc.3c07421>

Our proton release mechanistic proposal is shown in Appendix Fig. 10. We have now applied PROPKA to our simulations with inward-facing DgoT to analyze the role of conformational changes in changes in pKa of D46 and E133. We updated Figure 6, added a new Fig. EV4 and modified the manuscript accordingly (lines 243-255). Although these propKa calculations provide further support for the proton release pathway proposed here, we have also discussed the need to investigate alternative scenarios in lines 295-301, 430-434 and 511-520.

6. There is no figure in main text, like an introductory figure, showing the protein architecture, membrane around it and the location of the residues with respect to membrane. I think this should be provided for readers to get acquainted with the topic early on.

We now provide such information with the new Figure 1.

7. What do the dotted lines represent in Fig. 2d, 2e, etc?

The dotted lines indicate distances that correspond to open and closed conformations of the extracellular gate. We added this information to the caption of Figure 3 (updated number in the revised version of the manuscript).

8. E133Q structure was used to construct WT system. Authors must provide more details on how they relaxed the structure after converting mutant to WT.

The structure of the mutant was converted to WT prior to equilibration. We now added this information in Methods section (lines 701-703). The equilibration protocol was the same for all classical MD simulations (lines 725-733).

9. Galactonate charge/parameters must be provided in supplementary information for reproducibility.

We have deposited the galactonate parameters (protonated and deprotonated) on the following repository: <https://jugit.fz-juelich.de/n.dmitrieva/dgot-transport-mechanism/>

10. MD simulation methods - pKa calculations can be used to decipher ground state protonation states. Also, provide information if histidines were delta or epsilon protonated in their charge neutral states.

Since it was evident from structural and mutagenesis data that D46 and E133 are involved in proton transport, we intentionally simulated systems with all possible combinations of protonation states of these two residues, rather than concentrating on their ground state protonation states. The rest of the titratable residues (except for H56) are located far from the active center of the protein, and most of them are well solvated; we therefore chose standard protonation states. Since H56 forms a salt bridge with E180 in the inward crystal structure (PDB ID 6E9N), we decided to keep it protonated. This decision is supported by PROPKA pKa estimates for H56 of 6.38 and 6.18 in outward and inward crystal structures, respectively. The other three histidines (see Fig. EV1) were in the neutral charge state and epsilon protonated; we added this information in the Methods section (line 695).

11. Missing residues were modeled - how was the protein relaxed after modeling - can these affect authors results and conclusions?

Missing residues were modeled with the SWISS MODEL webserver and system equilibration procedures are described in Methods section (lines 696-699). Most of the missing residues that were modeled are located in the loops rather than in the transmembrane region; therefore we do not expect them to be critical for the protein transport activity. The only modeled fragment that is close to the protein core is part of TM7 (residues 278-286), which is resolved as a helix only in the inward-facing structure (PDB ID 6E9N), but missing in the outward-facing one (6E9O). Although this fragment is relatively close to the extracellular gate, we believe that modeling these residues in the outward-facing structure did not affect significantly its dynamics, since the hydrophobic belt controlling water access and substrate binding is formed by residues located in the crystallographically resolved part of TM7 (i.e. F274, L275 and W277).

12. The time step in QM/MM equilibration steps was small and also in production runs. Why is that? With larger time step, were there problems in simulations?

We apologize for the misunderstanding. All our QM/MM MD simulations were run in the Born-Oppenheimer (BO) scheme. During the initial QM/MM BO MD-based annealing and heating, we take the precaution of using a smaller timestep (10 a.u.). Afterwards, we normally increase the timestep (to 20 a.u.) for the QM/MM BOMD-based equilibration runs (as well as for enhanced sampling simulations, such as thermodynamic integration used here). We have used a similar protocol for previous QM/MM studies on ion channels and transporters (Chiariello et al. (2020). *J Am Chem Soc* **142**, 7254-7258; Chiariello et al. (2021). *J Phys*

Chem Lett, 4415-4420; Schackert et al. (2023). *J Chem Inf Model* **63**, 1293-1300). We have now clarified this point in the Methods on lines 802-815.

Further changes in the manuscript:

Figure 3: Views in panels A-C showing the arrangement of the charged residues were zoomed in to improve visibility.

Figure 8: During the curation of the source data we realized there was a typo on the code used to generate this plot. After correction, the curves corresponding to D46N in panel 8E (as well as fit parameters for this mutant in Table 1) and R47Q in panel 8H were updated.

Lines 260-261: When addressing the comments of reviewer 2, we realized that there was a typo in the barrier listed for the D46 side chain rotation, which has now been corrected.

Lines 789-790 and 792: Missing details on the basis set and link atoms used in the QM/MM calculations have been added.

We would like to thank all three reviewers for their helpful criticisms, and we are grateful for the opportunity to submit a revised version of our manuscript. We look forward to your decision regarding its publication in the *EMBO Journal*. Please contact me if we can provide additional information or if there are other concerns.

Sincerely yours,

Mercedes Alfonso Prieto, PhD
Group Leader
Institute of Neuroscience and Medicine
Computational Biomedicine (INM-9)
Forschungszentrum Jülich, Germany

Christoph Fahlke, MD
Professor
Institute of Biological Information
Processing
Molekular- und Zellphysiologie (IBI-1)
Forschungszentrum Jülich, Germany

Dear Prof. Fahlke,

Thank you submitting a revised version of your manuscript. It was sent to the same reviewers that originally appraised your work; we have now heard from two of them. Their comments are attached to the bottom of this email. As you will see, both are satisfied with the changes you made. Before we can move forwards towards publication of your manuscript, though, there are some remaining editorial points which need to be addressed. In this regard, would you please:

- mark the corresponding author in the manuscript file,
 - change the title of the 'Conflict of Interest' statement to the "Disclosure and competing interests statement",
 - remove the author credit section from the manuscript,
 - reduce the text in the last column of the checklist to only section name, no explanation is necessary if the response is "Not Applicable",
 - include the manuscript title on the title page of Appendix 1 (Appendix for Transport mechanism of DgoT, a bacterial homolog of SLC17 organic anion transporters),
 - define box plots in terms of minima, maxima, centre, bounds of box and whiskers, and percentile in the legends of figures 3d-e; 5d; 9a-b; EV 4b,
 - define n in the legends of figures 3d-e; 4c; 5c-d; 6b; 8b; 9a-b; EV 3c; EV 4b.
 - define error bars are not defined in the legends of figures 2c; 4c; 5c; 8c-f, h. Please note that n=2 in figure 8g. In this case the data range may be indicated with a bar,
 - ensure that legends for figures 8e-g are provided in a sequential manner (at present, the legend for figure 8f-g is provided before the legend of figure 8e),
- provide a legend for figure 6e and figure EV 2a-b, and
- state exact p values in the legend of figure 9e.

I look forward to receiving these changes. EMBO Press is an editorially independent publishing platform for the development of EMBO scientific publications.

Yours sincerely,

William Teale

William Teale, PhD
Editor
The EMBO Journal
w.teale@embojournal.org

Please remember: Digital image enhancement is acceptable practice, as long as it accurately represents the original data and

conforms to community standards. If a figure has been subjected to significant electronic manipulation, this must be noted in the figure legend or in the 'Materials and Methods' section. The editors reserve the right to request original versions of figures and the original images that were used to assemble the figure.

We realize that it is difficult to revise to a specific deadline. In the interest of protecting the conceptual advance provided by the work, we recommend a revision within 3 months (26th Dec 2024). Please discuss the revision progress ahead of this time with the editor if you require more time to complete the revisions. Use the link below to submit your revision:

Referee #1:

I see that the authors have reasonably addressed the reviewer's comments in the revised manuscript. I therefore believe that the manuscript is now suitable for publication.

Referee #3:

Authors have answered all my questions. I support the publication of the manuscript.

All editorial and formatting issues were resolved by the authors.

Dear Christoph,

I am pleased to inform you that your manuscript has been accepted for publication in the EMBO Journal.

Congratulations to you and your team!

Yours sincerely,

William

William Teale, PhD
Editor
The EMBO Journal
w.teale@embojournal.org
